# Intrinsic Benefits of Categorical Distributional Loss: Uncertainty-aware Regularized Exploration in Reinforcement Learning

**Ke Sun[1], Yingnan Zhao[2], Enze Shi[1], Yafei Wang[1], Xiaodong Yan[3], Bei Jiang[1], Linglong Kong[1]\***

[1]University of Alberta, Canada
[2] Harbin Engineering University, China
[3]Xi'an Jiaotong University, China
{ksun6,eshi,yafei2,bei1,lkong}@ualberta.ca
zhaoyingnan@hrbeu.edu.cn
yanxiaodong@xjtu.edu.cn

## Abstract

The remarkable empirical performance of distributional reinforcement learning (RL) has garnered increasing attention to understanding its theoretical advantages over classical RL. By decomposing the categorical distributional loss commonly employed in distributional RL, we find that the potential superiority of distributional RL can be attributed to a derived distribution-matching entropy regularization. This less-studied entropy regularization aims to capture additional knowledge of return distribution beyond only its expectation, contributing to an augmented reward signal in policy optimization. In contrast to the vanilla entropy regularization in MaxEnt RL, which explicitly encourages exploration by promoting diverse actions, the novel entropy regularization derived from categorical distributional loss implicitly updates policies to align the learned policy with (estimated) environmental uncertainty. Finally, extensive experiments verify the significance of this uncertainty-aware regularization from distributional RL on the empirical benefits over classical RL. Our study offers an innovative exploration perspective to explain the intrinsic benefits of distributional learning in RL.

## 1 Introduction

The fundamental characteristics of classical reinforcement learning (RL) [57], such as Q-learning [62], rely on estimating the expectation of discounted cumulative rewards that an agent observes while interacting with the environment. In contrast to the expectation-based RL, a novel branch of algorithms, termed *distributional RL*, seeks to estimate the entire distribution of total returns and has achieved state-of-the-art performance across a diverse array of environments [5, 11, 10, 68, 71, 43, 63, 56, 52]. Meanwhile, discussions of distributional RL have increasingly extended into a broader range of fields, such as risk-sensitive control [10, 31, 8], offline learning [35, 67], policy exploration [36, 49, 9, 25], robustness [55, 53, 51], optimization [51, 28, 54], statistical inference [69], multivariate rewards [70, 66], and continuous-time setting [65].

**Motivation: Understanding the Benefits of Employing (Categorical) Distributional Loss in RL.**
Despite the impressive empirical success of distributional RL algorithms, our comprehension of their advantages over classical RL remains incomplete, especially for the general function approximation setting and practical implementations. Early work [33] demonstrated that in many realizations of tabular and linear approximation settings, distributional RL behaves similarly to classic RL, suggesting

---

*Corresponding author

39th Conference on Neural Information Processing Systems (NeurIPS 2025).

that its benefits are mainly realized in the non-linear approximation setting. Although their findings offer profound insights, their analysis, based on a coupled update method, overlooks several factors, such as the optimization effect under various losses. The statistical benefits of quantile temporal difference (QTD), employed in quantile distributional RL, e.g., QR-DQN [11], were highlighted in [51, 50], which posited that the robust estimation of QTD fosters the benefits in stochastic environments. The foundational theoretical aspects of Categorical Distributional RL (CDRL), e.g., C51 [5], were discussed in [48, 27]; however, explaining the advantages of categorical distributional learning remains under-explored. Recent studies [60, 59] elucidate the benefits of distributional RL by introducing the small-loss and second-order PAC bounds, revealing the enhanced sample efficiency, particularly in specific cases with small achievable costs. Yet, their findings are not directly based on practical distributional RL algorithms, such as C51 or QR-DQN. Therefore, it is imperative to close this gap between understanding their theoretical advantages and practical deployment in complex environments for distributional RL algorithms.

**Contributions.** In this study, we interpret the potential superiority of distributional learning in RL over classical RL, specifically focusing on Categorical Distributional RL (CDRL), the pioneering family within distributional RL. We examine the benefits through the lens of a regularized exploration effect, offering a distinct perspective relative to existing literature. Our investigation begins by decomposing the categorical distributional loss into a mean-related term and a distribution-matching regularization term, facilitated by our proposed return density decomposition technique. The resulting regularization acts as an augmented reward in the actor critic framework, encouraging policies to explore states whose *current return distribution estimates lag far behind the (estimated) environmental uncertainty in the target return*. This derived regularization from the categorical distributional loss in CDRL promotes an uncertainty-aware exploration effect, which diverges from the exploration for diverse actions commonly used in MaxEnt RL [64, 17, 18]. We also provide the convergence foundations when leveraging the decomposed uncertain-aware regularization in the actor critic. Empirical evidence underscores the pivotal role of the uncertainty-aware entropy regularization in the empirical success of adopting categorical distributional loss in RL over classical RL on both Atari games and MuJoCo tasks. We further elucidate the distinct roles that the uncertainty-aware entropy in distributional RL and the vanilla entropy in MaxEnt RL play by exploring their mutual impacts on learning performance, providing consitent evidence for []. This opens new avenues for future research in this domain. Our contributions are summarized as follows:

1. By applying a return density decomposition on the categorical distributional loss, we derive a distribution-matching regularization. This regularization promotes uncertainty-aware exploration, interpreting the benefits of categorical distributional learning in RL.

2. We extend the benefit interpretation of the categorical distributional loss to policy gradient methods. We compare the different exploration effects of our decomposed uncertainty-aware regularization from distributional RL and the vanilla entropy regularization in MaxEnt RL.

3. Empirically, we verify the uncertainty-aware regularization effect on the performance improvement of distributional RL and investigate the mutual impacts of regularizations in the learning.

**Outline.** We provide the related work and background knowledge in Sections 2 and 3, respectively. We begin by interpreting the benefits of categorical distributional learning as uncertainty-aware regularized exploration in value-based RL in Section 4. We further probe this exploration benefit in the policy-based RL, especially the actor critic framework, in Section 5, where we directly compare it with the vanilla entropy regularization in MaxEnt RL. Extensive experiments demonstrate the benefits of regularized exploration in distributional RL and its mutual impact with entropy regularization in MaxEnt RL in Section 6.

## 2   Related Work

**Distributional Learning via Categorical Representation.** Categorical learning has been widely employed, with advantages in representation [46, 26] and optimization [24, 54]. Recently, the empirical superiority of categorical distribution learning has been further investigated in various RL tasks [14]. A pressing need exists to examine the theoretical foundations of categorical distributional learning, particularly in RL. The perspective of uncertainty-aware regularized exploration that our study introduces provides significant insights into understanding the benefits of employing categorical distribution loss in the RL context.

**Uncertainty-oriented Exploration.** Uncertainty-oriented exploration plays an integral part in existing exploration methods [20], which leverages uncertainty either in the (posterior) estimation of the value function, as seen in Bayesian framework [45, 4], Bootstrap [44], and Ensemble methods [30], or in the entire distribution of returns [58, 36, 9]. For example, Decaying Left Truncated Variance (DLTV) [36] and Perturbed Quantile Regression (PQR) [9] exploit the variability of the learned return distribution to promote an optimistic exploration in distributional RL. In contrast, the primary aim of this study is to demonstrate that distributional learning in RL entails an intrinsic exploration effect against environmental uncertainty, contributing to the outperformance of distributional RL over classical RL. Our study goal is independent of designing advanced exploration strategies on top of distributional RL. Similarly, MaxEnt RL [64], which includes soft Q-learning [16], Soft Actor Critic (SAC) [17] and their variants [19], also promotes uncertainty-oriented exploration by relying on the stochasticity of the learned policy. A more detailed discussion of related work is provided in Appendix A.

## 3 Preliminaries

**Markov Decision Process (MDP) and Classical RL.** An environment is modeled via an Markov Decision Process $(\mathcal{S}, \mathcal{A}, \mathcal{R}, P, \gamma)$, with a set of states $\mathcal{S}$ and actions $\mathcal{A}$, the bounded reward function $\mathcal{R} : \mathcal{S} \times \mathcal{A} \to \mathcal{P}([R_{\min}, R_{\max}])$, the transition kernel $P : \mathcal{S} \times \mathcal{A} \to \mathcal{P}(\mathcal{S})$, and a discounted factor $\gamma \in [0, 1]$. We denote the reward the agent receives at time $t$ as $r_t \sim \mathcal{R}(s_t, a_t)$. Given a policy $\pi$, the key quantity of interest is the return $Z^\pi$, which is the total cumulative rewards over the course of a trajectory defined by $Z^\pi(s, a) = \sum_{t=0}^\infty \gamma^t r_t | s_0 = s, a_0 = a$. Classical RL focuses on estimating the expectation of the return, i.e., $Q^\pi(s, a) = \mathbb{E}_\pi \left[ \sum_{t=0}^{+\infty} \gamma^t r_t | s_0 = s, a_0 = a \right]$. We also define Bellman evaluation operator $\mathcal{T}^\pi Q(s, a) = \mathbb{E}[\mathcal{R}(s, a)] + \gamma \mathbb{E}_{s' \sim P, a' \sim \pi} [Q(s', a')]$, and Bellman optimality operator $\mathcal{T}^{\text{opt}} Q(s, a) = \mathbb{E}[\mathcal{R}(s, a)] + \gamma \max_{a'} \mathbb{E}_{s' \sim P} [Q(s', a')]$.

**Distributional RL and CDRL.** Instead of only learning the expectation in classical RL, distributional RL models the full distribution of the return $Z^\pi$. The return distribution $\eta^\pi : \mathcal{S} \times \mathcal{A} \to \mathcal{P}(\mathbb{R})$ is defined as $\eta^\pi(s, a) = \mathcal{D}(Z^\pi(s, a))$, where $\mathcal{D}$ extracts the distribution of a random variable. $\eta^\pi(s, a)$ is updated via the distributional Bellman operator $\mathfrak{T}^\pi$, defined by $\mathfrak{T}^\pi Z(s, a) \stackrel{D}{=} \mathcal{R}(s, a) + \gamma Z(s', a')$, where $\stackrel{D}{=}$ implies that random variables of both sides are equal in distribution. Categorical Distributional RL (CDRL) is the first successful distributional RL family that approximates the return distribution by a discrete categorical distribution $\widehat{\eta}^\pi = \sum_{i=1}^N p_i \delta_{z_i}$, where $\{z_i\}_{i=1}^N$ is a set of fixed supports and $\{p_i\}_{i=1}^N$ are learnable probabilities. The leverage of a heuristic projection operator $\Pi_{\mathcal{C}}$ (see Appendix B for more details) and the Kullback–Leibler (KL) divergence guarantee the theoretical convergence of CDRL under Cramér distance or Wasserstein distance in the tabular setting [48].

## 4 Regularization Benefits in Value-based Distribution RL

In this section, we simplify value-based distributional RL to a Neural Fitted Z-Iteration (Neural FZI) process in Section 4.1, within which the distributional loss used in distributional RL can be further rewritten as an entropy-regularized form as shown in Section 4.2. Finally, we characterize the role of the derived entropy-based regularization as uncertain-aware regularized exploration in Section 4.3.

### 4.1 Distributional RL: Neural FZI

**Classical RL: Neural Fitted Q-Iteration (Neural FQI).** Neural FQI [13, 47] offers a statistical explanation of DQN [39], capturing its key features, including experience replay and the target network $Q_{\theta^*}$. In Neural FQI, we update a parameterized $Q_\theta$ in each iteration $k$ of an iterative regression framework:

$$Q_\theta^{k+1} = \operatorname{argmin}_{Q_\theta} \frac{1}{n} \sum_{i=1}^n \left[ y_i^k - Q_\theta(s_i, a_i) \right]^2, \tag{1}$$

where the target $y_i^k = r(s_i, a_i) + \gamma \max_{a \in \mathcal{A}} Q_{\theta^*}^k(s_i', a)$ is fixed within every $T_{\text{target}}$ steps to update target network $Q_{\theta^*}$ by letting $Q_{\theta^*}^k = Q_\theta^k$. The experience buffer induces independent samples

$\{(s_i, a_i, r_i, s'_i)\}_{i \in [n]}$. If $\{Q_\theta : \theta \in \Theta\}$ is sufficiently large such that it contains $\mathcal{T}^{\text{opt}}Q_{\theta*}^k$, i.e., the realizable assumption in learning theory [40], Neural FQI has the solution $Q_\theta^{k+1} = \mathcal{T}^{\text{opt}}Q_{\theta*}^k$, which is exactly the updating rule under Bellman optimality operator [13].

**Distributional RL: Neural Fitted Z-Iteration (Neural FZI).** Analogous to Neural FQI, we simplify value-based distributional RL algorithms with the parameterized $Z_\theta$ as Neural FZI:

$$Z_\theta^{k+1} = \underset{Z_\theta}{\text{argmin}} \frac{1}{n} \sum_{i=1}^{n} d_p(Y_i^k, Z_\theta(s_i, a_i)), \tag{2}$$

where we denote the target return as $Y_i^k = \mathcal{R}(s_i, a_i) + \gamma Z_{\theta*}^k(s'_i, \pi_Z(s'_i))$ with the policy $\pi_Z$ following the greedy rule $\pi_Z(s'_i) = \text{argmax}_{a'} \mathbb{E}\left[Z_{\theta*}^k(s'_i, a')\right]$. The target $Y_i^k$ is fixed within every $T_{\text{target}}$ steps to update target network $Z_{\theta*}$. $d_p$ is a distribution divergence between two distributions. While our analysis is not intended to involve properties of deep neural networks, we interpret distributional RL as Neural FZI, as it is by far the closest to the practical algorithms. More details about the motivation of Neural FZI are provided in Appendix C.

## 4.2 Distributional RL: Entropy-regularized Neural FQI

As mentioned previously in preliminary knowledge in Section 3, CDRL employs neural networks to learn the probabilities $\{p_i\}_{i=1}^N$ in a discrete categorical distribution to represent $Z_\theta$, and choose KL divergence as $d_p$ in Eq. 2 of Neural FZI. We next decompose the KL-based distributional loss $d_p$ in CDRL by utilizing an equivalent histogram density estimator $\widehat{p}$ in representing $Z_\theta$.

**Return Density Decomposition.** To characterize the impact of additional knowledge from the return distribution beyond its expectation, we use a variant of *gross error model* from robust statistics [23], which was also similarly applied to analyze Label Smoothing [42] and Knowledge Distillation [21]. Akin to the categorical parameterization in CDRL, we utilize a histogram function estimator $\widehat{p}^{s,a}(x)$

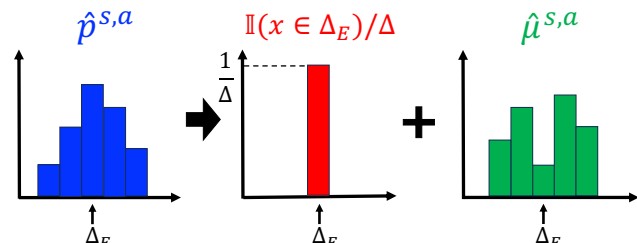

Figure 1: Return Density Decomposition on Histograms.

with $N$ bins to approximate an arbitrary continuous density $p^{s,a}(x)$ of $Z^\pi(s, a)$, given a state $s$ and action $a$. In contrast to categorical parameterization defined on a set of fixed supports, the histogram estimator operates over a continuous interval, enabling more nuanced analysis within continuous functions. Given a fixed set of supports $l_0 \leq l_1 \leq ... \leq l_N$ with the equal bin size as $\Delta$, each bin is thus denoted as $\Delta_i = [l_{i-1}, l_i), i = 1, ..., N-1$ with $\Delta_N = [l_{N-1}, l_N]$. As such, the histogram density estimator is formulated by $\widehat{p}^{s,a}(x) = \sum_{i=1}^N p_i \mathbb{1}(x \in \Delta_i)/\Delta$ with $p_i$ as the coefficient in the $i$-th bin $\Delta_i$. Denote $\Delta_E$ as the interval that $\mathbb{E}[Z^\pi(s, a)]$ falls into, i.e., $\mathbb{E}[Z^\pi(s, a)] \in \Delta_E$. Putting all together, we apply an action-state return density decomposition over the histogram density estimator $\widehat{p}^{s,a}$:

$$\widehat{p}^{s,a}(x) = (1 - \epsilon)\mathbb{1}(x \in \Delta_E)/\Delta + \epsilon\widehat{\mu}^{s,a}(x), \tag{3}$$

where $\widehat{p}^{s,a}$ is decomposed into a single-bin histogram $\mathbb{1}(x \in \Delta_E)/\Delta$ with all mass on $\Delta_E$ and an **induced** histogram density function $\widehat{\mu}^{s,a}$ evaluated by $\widehat{\mu}^{s,a}(x) = \sum_{i=1}^N p_i^\mu \mathbb{1}(x \in \Delta_i)/\Delta$ with $p_i^\mu$ as the coefficient of the $i$-th bin $\Delta_i$. $\epsilon$ is a hyper-parameter pre-specified before the decomposition, controlling the proportion between $\mathbb{1}(x \in \Delta_E)/\Delta$ and $\widehat{\mu}^{s,a}(x)$. See Figure 1 for the illustration of the decomposition. More specifically, the induced histogram density function $\widehat{\mu}^{s,a}$ in the second term of Eq. 3 represents the difference between the full histogram function $\widehat{p}^{s,a}$ and a single-bin histogram $\mathbb{1}(x \in \Delta_E)/\Delta$, where $\mathbb{1}(x \in \Delta_E)/\Delta$ only captures the mean. This difference indicates that $\widehat{\mu}^{s,a}$ *captures the additional distribution information of $Z^\pi(s, a)$ beyond its expectation* $\mathbb{E}[Z^\pi(s, a)]$, *incorporating higher-moments information*. This reflects the influence of using a full distribution on the performance of distributional RL. The additional leverage of $\widehat{\mu}^{s,a}$ in the distributional loss explains the behavior differences between classical and distribution RL algorithms. We next demonstrate that $\widehat{\mu}^{s,a}$ is a valid probability density under certain $\epsilon$ in Proposition 1.

**Proposition 1.** *(Decomposition Validity) Denote* $\widehat{p}^{s,a}(x \in \Delta_E) = p_E \frac{\mathbb{1}(x \in \Delta_E)}{\Delta}$, *where* $p_E$ *is the coefficient on the bin* $\Delta_E$. $\widehat{\mu}^{s,a}(x) = \sum_{i=1}^{N} p_i^{\mu} \mathbb{1}(x \in \Delta_i)/\Delta$ *is a valid density if and only if* $\epsilon \geq 1 - p_E$.

The proof can be found in Appendix D. Proposition 1 demonstrates that the return density decomposition is valid when the hyper-parameter $\epsilon$ is well specified as $\epsilon \geq 1 - p_E$. Under this condition, our analysis maintains the standard categorical distributional learning in distributional RL.

**Equivalence between Histogram Parameterization and Categorical Representation.** The histogram function is a continuous estimator in contrast to the discrete nature of categorical parameterization. Although the underlying connection is relatively straightforward, we still demonstrate their equivalence in representing a density function in Appendix E for completeness. As a supplementary analysis, with attribution to [61], we also discuss the necessary theoretical underpinnings of the histogram density estimator in the context of distributional RL in Appendix F.

**Distributional RL: Entropy-regularized Neural FQI.** We apply the decomposition in Eq. 3 on the histogram density function, denoted as $\widehat{p}^{s_i', \pi_Z(s_i')}$, of the target return $Y_i^k = \mathcal{R}(s_i, a_i) + \gamma Z_{\theta^*}^k(s_i', \pi_Z(s_i'))$ in Eq. 2 of Neural FZI. Consequently, we have $\widehat{p}^{s_i', \pi_Z(s_i')}(x) = (1 - \epsilon)\mathbb{1}(x \in \Delta_E^i)/\Delta + \epsilon\widehat{\mu}^{s_i', \pi_Z(s_i')}(x)$, where $\Delta_E^i$ represents the interval that the expectation of the target return $Y_i^k$ falls into, i.e., $\mathbb{E}\left[Y_i^k\right] \in \Delta_E^i$, and $\widehat{\mu}^{s_i', \pi_Z(s_i')}$ is the induced histogram density function, similar to the role of $\widehat{\mu}^{s,a}$ in Eq. 3. Let $\mathcal{H}(U, V)$ be the cross-entropy between two probability measures $U$ and $V$, i.e., $\mathcal{H}(U, V) = -\int_{x \in \mathcal{X}} U(x) \log V(x) \, \mathrm{d}x$. Immediately, we can derive the following entropy-regularized loss function form of Neural FZI for distributional RL in Proposition 2. The proof is provided in Appendix H.

**Proposition 2.** *(Decomposed Neural FZI) Denote* $q_{\theta}^{s,a}$ *as the histogram density estimator of* $Z_{\theta}^k(s, a)$ *in Neural FZI. Based on the decomposition in Eq. 3 and the KL divergence as* $d_p$, *the Neural FZI process in Eq. 2 is simplified as*

$$Z_{\theta}^{k+1} = \underset{q_\theta}{\operatorname{argmin}} \frac{1}{n} \sum_{i=1}^{n} [\underbrace{-\log q_{\theta}^{s_i, a_i}(\Delta_E^i)}_{\text{Mean-Related Term}} + \underbrace{\alpha\mathcal{H}(\widehat{\mu}^{s_i', \pi_Z(s_i')}, q_{\theta}^{s_i, a_i})}_{\text{Regularization Term}}], \tag{4}$$

*where* $\alpha = \varepsilon/(1 - \varepsilon) > 0$ *and the mean-related term is negative log-likelihood centered on* $\Delta_E^i$.

**Connection between Neural FQI and FZI.** A crucial bridge between classical RL and distributional RL is established in Proposition 3, where we demonstrate that minimizing the mean-related term in Eq. 29 of Neural FZI is asymptotically equivalent to minimizing Neural FQI in terms of the minimizers as $\Delta \to 0$. As such, with this equivalence in the objective function, the remaining regularization term $\alpha\mathcal{H}(\widehat{\mu}^{s_i', \pi_Z(s_i')}, q_{\theta}^{s_i, a_i})$ in Eq. 29 thus interprets the potential benefits of CDRL over classical RL. For the uniformity of notation, we still use $s, a$ in the following analysis instead of $s_i, a_i$.

**Proposition 3.** *(Equivalence between **the Mean-Related term** in Decomposed Neural FZI and Neural FQI) In Eq. 29, assume the function class* $\{Z_\theta : \theta \in \Theta\}$ *is sufficiently large such that it contains the target* $\{Y_i^k\}_{i=1}^n$ *for all k, when* $\Delta \to 0$, *minimizing **the mean-related term** in Eq. 29 implies*

$$\mathbb{P}(Z_{\theta}^{k+1}(s, a) = \mathcal{T}^{opt} Q_{\theta^*}^k(s, a)) = 1, \tag{5}$$

*where* $\mathcal{T}^{opt} Q_{\theta^*}^k(s, a)$ *is the scalar-valued target in the k-th phase of Neural FQI.*

Proposition 3 demonstrates that as $\Delta \to 0$, the random variable $Z_{\theta}^{k+1}(s, a)$ with the limiting distribution in Neural FZI (distributional RL) will *degrade* to a constant $\mathcal{T}^{\text{opt}} Q_{\theta^*}^k(s, a)$, the minimizer (scalar-valued target) in Neural FQI (classical RL). That being said, *minimizing the mean-related term in Neural FZI is asymptotically equivalent to minimizing Neural FQI with the same limiting minimizer*. A formal proof for convergence in distribution with the convergence rate $o(\Delta)$ is given in Appendix I. The realizable assumption that $\{Z_\theta : \theta \in \Theta\}$ is sufficiently large such that it contains $\{Y_i^k\}_{i=1}^n$ implies good in-distribution generalization performance in each phase of Neural FZI, which is also adopted in [67]. This connection is also consistent with the mean-preserving property of distributional RL in the tabular setting [48], but we extend this conclusion to the arbitrary function approximation with a histogram density estimator. Proposition 3 especially focuses on the asymptotic property of the

mean-related term, which is different from existing convergence results based on the entire categorical distribution [48, 6]. Given the connection between optimizing the mean-related term of Neural FZI with Neural FQI in Proposition 3, we can leverage the regularization term $\alpha\mathcal{H}(\widehat{\mu}^{s_i', \pi_Z(s_i')}, q_\theta^{s_i, a_i})$ to explain the behavior difference between CDRL and classical RL, as analyzed later.

### 4.3 Uncertainty-aware Regularized Exploration

**Regularization Effect.** It turns out that minimizing the regularization term $\alpha\mathcal{H}(\widehat{\mu}^{s_i', \pi_Z(s_i')}, q_\theta^{s_i, a_i})$ in Neural FZI pushes $q_\theta^{s, a}$ for the current return density estimator to catch up with the target return density function of $\widehat{\mu}^{s_i', \pi_Z(s_i')}$. Importantly, $\widehat{\mu}^{s_i', \pi_Z(s_i')}$ encompasses the uncertainty of the entire return distribution in the learning course beyond only its expectation, given that $\widehat{\mu}^{s_i', \pi_Z(s_i')}$ is the induced histogram density after applying the return density decomposition in Eq. 3. Since it is a prevalent notion that distributional RL can significantly reduce intrinsic uncertainty of the environment [36, 10], the derived distribution-matching regularization term $\alpha\mathcal{H}(\widehat{\mu}^{s_i', \pi_Z(s_i')}, q_\theta^{s_i, a_i})$ helps to capture more uncertainty of the environment by modeling higher moments of the whole return distribution beyond the expectation. In Section 5, we further demonstrate that this derived regularization contributes to an *uncertainty-aware regularized exploration* effect in the policy optimization or actor critic.

**Remark: Approximation of $\widehat{\mu}^{s', \pi_Z(s')}$.** In practical distributional RL algorithms, we typically use temporal-difference (TD) learning to attain the target probability density estimate $\widehat{\mu}^{s', \pi_Z(s')}$ based on Eq. 3, provided $\mathbb{E}\left[Z(s, a)\right]$ exists and $\epsilon \geq 1 - p_E$ in Proposition 1. The approximation error of $\widehat{\mu}^{s', \pi_Z(s')}$ is fundamentally determined by the TD learning nature. A desirable approximation of $\widehat{\mu}^{s', \pi_Z(s')}$ intuitively leads to performance improvement in distributional RL. As KL divergence is used in CDRL, we also discuss the usage of KL divergence in distributional RL in Appendix G.

## 5 Regularization Benefits in Actor Critic

**Notations.** In this section, we use boldface notations to represent random variables, such as $(\mathbf{s}_t, \mathbf{a}_t)$, at time $t$ for clarity in the learning process of the actor critic.

### 5.1 Connection with MaxEnt RL

**Motivation for the Connection.** The maximum entropy regularization is commonly used in RL, which has various conceptual and practical advantages. Firstly, the learned policy is encouraged to visit states with high entropy in the future, promoting the exploration of diverse actions [19, 17, 64]. It also considerably improves the learning speed [37] and therefore is widely employed in state-of-the-art algorithms, e.g., Soft Actor-Critic (SAC) [17]. Similar empirical benefits of both distributional RL and MaxEnt RL motivate us to probe their underlying connection, especially by comparing their exploration effects.

**Explicit Entropy Regularization in MaxEnt RL.** MaxEnt RL *explicitly* encourages exploration by optimizing for policies to reach states with higher entropy in the future:

$$J(\pi) = \sum_{t=0}^{T} \mathbb{E}_{(\mathbf{s}_t, \mathbf{a}_t) \sim \rho_\pi} \left[ r\left(\mathbf{s}_t, \mathbf{a}_t\right) + \beta\mathcal{H}(\pi(\cdot|\mathbf{s}_t)) \right], \tag{6}$$

where $\mathcal{H}\left(\pi_\theta\left(\cdot|\mathbf{s}_t\right)\right) = -\sum_a \pi_\theta\left(a|\mathbf{s}_t\right) \log \pi_\theta\left(a|\mathbf{s}_t\right)$ and $\rho_\pi$ is the generated distribution following $\pi$. The temperature parameter $\beta$ determines the relative importance of the entropy term against the cumulative rewards and thus controls the action diversity of the optimal policy learned via Eq. 6.

**Implicit Entropy Regularization in Distributional RL.** For a direct comparison with MaxEnt RL, it is required to specifically analyze the impact of the regularization term in Eq. 29. Therefore, we directly incorporate the distribution-matching regularization of distributional RL in Eq. 29 into the Actor Critic (AC) framework, enabling us to consider a new soft Q-value. The new Q function can be computed iteratively by applying a modified Bellman operator denoted as $\mathcal{T}_d^\pi$, called *Distribution-Entropy-Regularized Bellman Operator*. Given a fixed $q_\theta$, $\mathcal{T}_d^\pi$ is defined as

$$\mathcal{T}_d^\pi Q\left(\mathbf{s}_t, \mathbf{a}_t\right) \triangleq r\left(\mathbf{s}_t, \mathbf{a}_t\right) + \gamma\mathbb{E}_{\mathbf{s}_{t+1} \sim P(\cdot|\mathbf{s}_t, \mathbf{a}_t)} \left[V\left(\mathbf{s}_{t+1}\right)\right], \tag{7}$$

where a new soft value function $V\left(\mathbf{s}_t\right)$ is defined by

$$V\left(\mathbf{s}_t\right) = \mathbb{E}_{\mathbf{a}_t \sim \pi} \left[Q\left(\mathbf{s}_t, \mathbf{a}_t\right) + f(\mathcal{H}\left(\mu^{\mathbf{s}_t, \mathbf{a}_t}, q_\theta^{\mathbf{s}_t, \mathbf{a}_t}\right))\right], \tag{8}$$

where $f$ is a continuous increasing function over the cross-entropy $\mathcal{H}$. $\mu^{\mathbf{s}_t,\mathbf{a}_t}$ is the induced true target return histogram density function via the decomposition in Eq. 3, which excludes its expectation. Note that $\mu^{\mathbf{s}_t,\mathbf{a}_t}$ can be approximated via bootstrap TD estimate $\widehat{\mu}^{\mathbf{s}_{t+1},\pi_Z(\mathbf{s}_{t+1})}$ similar to Eq. 29. In this specific tabular setting regarding $\mathbf{s}_t, \mathbf{a}_t$, we particularly use $q_\theta^{\mathbf{s}_t,\mathbf{a}_t}$ to approximate the true density function of $Z(\mathbf{s}_t, \mathbf{a}_t)$. The $f$ transformation over the

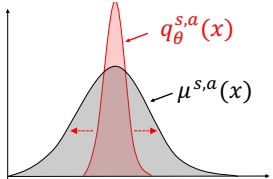 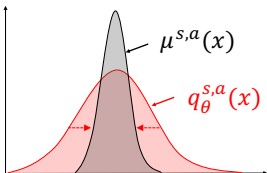

Figure 2: $q_\theta^{s,a}$ is optimized to disperse (left) or concentrate (right) to align with the uncertainty of target return distributions.

cross-entropy $\mathcal{H}$ between $\mu^{\mathbf{s}_t,\mathbf{a}_t}$ and $q_\theta^{\mathbf{s}_t,\mathbf{a}_t}(x)$ serves as the uncertainty-aware entropy regularization that we implicitly derive from value-based distributional RL in Section 4.2. By optimizing $q_\theta$ that is involved in the value-based critic component in actor critic, this regularization reduces the mismatch between the target return distribution and current estimate, aligning with the regularization effect analyzed in Section 4.3. As illustrated in Figure 2, $q_\theta^{s,a}$ is optimized to **catch up with** the uncertainty involved in the target return distribution of $\mu^{s,a}$, iteratively expanding the agent's knowledge about the environment uncertainty to contribute to more informative decisions. Next, we elaborate on its additional impact on policy learning in the actor critic compared to MaxEnt RL.

**Reward Augmentation for Policy Learning.** As opposed to the vanilla entropy regularization in MaxEnt RL that explicitly encourages the policy to explore, our derived regularization term in the distributional loss of RL plays the role of reward augmentation for policy learning. Compared with classical RL, the augmented reward from the distributional loss incorporates additional knowledge of the return distribution in the learning process. As we will show later, *the augmented reward encourages policies to reach states $\mathbf{s}_t$ with actions $\mathbf{a}_t \sim \pi(\cdot|\mathbf{s}_t)$, whose current action-state return distribution $q_\theta^{\mathbf{s}_t,\mathbf{a}_t}$ **lags far behind** the (estimated) environmental uncertainty from the target returns.*

For a detailed comparison with MaxEnt RL, we now focus on the properties of our decomposed distribution-matching regularization in the actor critic. In Lemma 1, we demonstrate that Distribution-Entropy-Regularized Bellman operator $\mathcal{T}_d^\pi$ inherits the convergence property in the policy evaluation phase with a cumulative augmented reward function as the new objective function $J'(\pi)$.

**Lemma 1.** *(Distribution-Entropy-Regularized Policy Evaluation) Consider the distribution-entropy-regularized Bellman operator $\mathcal{T}_d^\pi$ in Eq. 7 and assume $\mathcal{H}(\mu^{\mathbf{s}_t,\mathbf{a}_t}, q_\theta^{\mathbf{s}_t,\mathbf{a}_t})$ is bounded for all $(\mathbf{s}_t, \mathbf{a}_t) \in \mathcal{S} \times \mathcal{A}$. We define $Q^{k+1} = \mathcal{T}_d^\pi Q^k$. Given $q_\theta$, $Q^{k+1}$ will converge to a corrected Q-value of $\pi$ as $k \to \infty$ with the new objective function $J'(\pi)$ defined as*

$$J'(\pi) = \sum_{t=0}^{T} \mathbb{E}_{(\mathbf{s}_t,\mathbf{a}_t)\sim\rho_\pi} \left[ r\left(\mathbf{s}_t, \mathbf{a}_t\right) + \gamma f(\mathcal{H}\left(\mu^{\mathbf{s}_t,\mathbf{a}_t}, q_\theta^{\mathbf{s}_t,\mathbf{a}_t}\right)) \right]. \tag{9}$$

The updating rule is $\pi_{\text{new}} = \arg\max_{\pi' \in \Pi} \mathbb{E}_{\mathbf{a}_t \sim \pi'} \left[Q^{\pi_{\text{old}}}(\mathbf{s}_t, \mathbf{a}_t) + f(\mathcal{H}\left(\mu^{\mathbf{s}_t,\mathbf{a}_t}, q_\theta^{\mathbf{s}_t,\mathbf{a}_t}\right))\right]$ in phase of policy optimization. Next, we derive a new policy iteration algorithm, called *Distribution-Entropy-Regularized Policy Iteration (DERPI)*, alternating between policy evaluation and policy improvement. It provably converges to a policy regularized by the distribution-matching term.

**Theorem 1.** *(Distribution-Entropy-Regularized Policy Iteration) Repeatedly applying distribution-entropy-regularized policy evaluation in Eq. 7 and the policy improvement, the policy converges to an optimal policy $\pi^*$ such that $Q^{\pi^*}\left(\mathbf{s}_t, \mathbf{a}_t\right) \geq Q^\pi\left(\mathbf{s}_t, \mathbf{a}_t\right)$ for all $\pi \in \Pi$.*

Please refer to Appendix J for the proof of Lemma 1 and Theorem 1. Theorem 1 demonstrates that if we incorporate the decomposed regularization into the actor critic in Eq. 9, we can design a variant of "soft policy iteration" [17] that can guarantee the convergence to an optimal policy given any fixed $q_\theta$. In summary, our theoretical investigation is a variant of the standard analytical framework in MaxEnt RL that allows a comparable analysis. Importantly, we next recognize a fundamental difference between our decomposed entropy regularization and the vanilla entropy regularization in MaxEnt RL.

**Uncertainty-aware Regularized Exploration in CDRL Compared with MaxEnt RL.** For the objective function $J(\pi)$ in Eq. 6 of MaxEnt RL, the state-wise entropy $\mathcal{H}(\pi(\cdot|\mathbf{s}_t))$ is maximized explicitly *w.r.t.* $\pi$ for policies with a higher entropy in terms of diverse actions to encourage an explicit exploration. For the objective function $J'(\pi)$ in Eq. 9 of distributional RL, the policy $\pi$ is implicitly optimized through **the action selection process** $\mathbf{a}_t \sim \pi(\cdot|\mathbf{s}_t)$ guided by an augmented reward signal from the distribution-matching regularization $f(\mathcal{H}\left(\mu^{\mathbf{s}_t,\mathbf{a}_t}, q_\theta^{\mathbf{s}_t,\mathbf{a}_t}\right))$. Concretely, the

learned policy is encouraged to visit state $\mathbf{s}_t$ along with the policy-determined action via $\mathbf{a}_t \sim \pi(\cdot|\mathbf{s}_t)$, whose current action-state return distributions $q_\theta^{\mathbf{s}_t,\mathbf{a}_t}$ *lag far behind* the target return distributions with a large discrepancy. This discrepancy is measured by the magnitude of the cross entropy between two return distributions of $q_\theta^{\mathbf{s}_t,\mathbf{a}_t}$ and $\mu^{\mathbf{s}_t,\mathbf{a}_t}$. A large discrepancy indicates that the uncertainty of the current return distribution is considerably misestimated for the considered states, enabling an uncertainty-aware exploration against these states in the policy optimization phase. This also indicates that the policy learning in CDRL is additionally driven by the uncertainty difference between the current and the target estimates, leading to a distinct exploration strategy of distributional RL relative to MaxEnt RL.

**Interplay of Uncertainty-aware Regularization in Distributional Actor Critic.** Putting the critic and actor learning together in distributional RL, we reveal their interplay impact pertinent to the uncertainty-aware regularized exploration. For the actor component, the policy learning seeks states and actions whose current return distribution estimate lags far behind the environmental uncertainty of the target returns. For the critic component, the critic learning reduces the return distribution mismatch on the states and actions explored by the policy, with two situations illustrated in Figure 2. This uncertainty-aware exploration effect arises from the decomposed regularization via the return density decomposition, interpreting the benefits of CDRL over classical RL.

## 6 Experiments

We comprehensively demonstrate our theoretical analysis using both Atari games and MuJoCo tasks. In Section 6.1, we validate that the uncertainty-aware regularization is crucial to the outperformance of CDRL over classical RL by varying $\epsilon$ in the return density decomposition. We also investigate the mutual impacts between the vanilla entropy regularization in MaxEnt RL and the uncertainty-aware entropy regularization from CDRL in Section 6.2. More implementation details, including the description of baselines, are provided in Appendix K.

### 6.1 Regularization Effect in Performance by Varying $\epsilon$

**Baseline Algorithm:** $\mathcal{H}(\mu, q_\theta)(\varepsilon = 0.8/0.5/0.1)$. For the categorical distributional loss in C51 or the distributional critic loss in the actor critic, we employ $\widehat{\mu}^{s,a}$ instead of $\widehat{p}^{s,a}$ as the target return distribution, leading to the decomposed algorithms, denoted by $\mathcal{H}(\mu, q_\theta)$. This decomposed algorithm enables us to assess the uncertainty-aware regularization effect of distributional RL by directly comparing its performance with the classical RL and CDRL.

**Experimental Details.** We substantiate that the decomposed uncertainty-aware entropy regularization, derived in Eq. 29 through the return density function decomposition, plays a crucial role in the empirical superiority of CDRL over classical RL. We compare CDRL with the decomposed baseline algorithm $\mathcal{H}(\mu, q_\theta)$ under different $\epsilon$ based on Eq. 3. To ensure a pre-specified $\epsilon$ that guarantees a valid decomposition analyzed in Proposition 1, we employ a new notation $\varepsilon$, which is proportional to

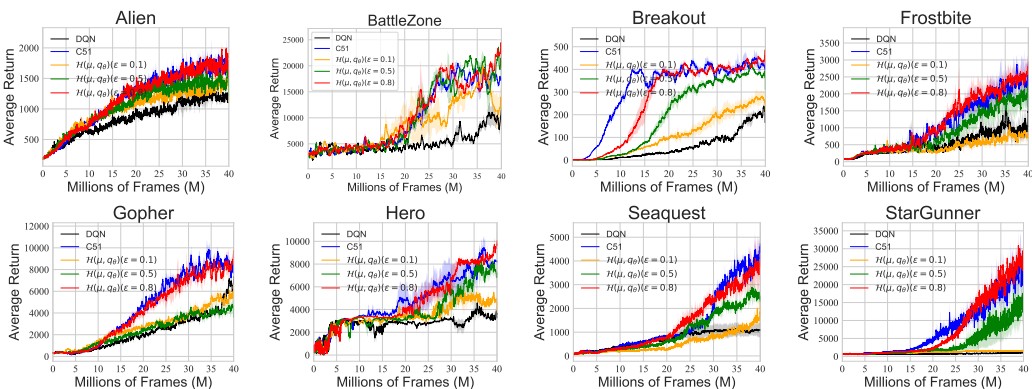

Figure 3: Learning curves of value-based CDRL (C51) and the decomposed algorithm $\mathcal{H}(\mu, q_\theta)(\varepsilon = 0.8/0.5/0.1)$ after applying the return distribution decomposition with different $\varepsilon$ on eight Atari games. Results are averaged over three seeds, and the shade represents the standard deviation.

$\epsilon$ but is more convenient in the implementation. See Appendix K.2 for more explanation, including the transformation equation between $\epsilon$ and $\varepsilon$, and the details of the baseline algorithm $\mathcal{H}(\mu, q_\theta)$.

**Results.** Figure 3 showcases that as $\varepsilon$ gradually decreases from 0.8 to 0.1, learning curves of decomposed C51, i.e., $\mathcal{H}(\mu, q_\theta)(\varepsilon = 0.8/0.5/0.1)$, tend to degrade from C51 to DQN across most Atari games. The sensitivity of the decomposed algorithm $\mathcal{H}(\mu, q_\theta)$ regarding $\varepsilon$ depends on the environment. Similar results in MuJoCo environments can be found in Appendix L.1. Overall, our empirical result corroborates that the decomposed uncertainty-aware entropy regularization from the categorical distributional loss is pivotal to the empirical advantage of CDRL over classical RL.

### 6.2 Mutual Impacts of the Two Entropy Regularization

**Baseline Algorithms.** For a detailed comparison of the mutual impacts between **V**anilla **E**ntropy (**VE**) in MaxEnt RL and **U**ncertainty-aware **E**ntropy (**UE**) in CDRL, we conduct an ablation study across several related baseline algorithms. We denote SAC with/without vanilla entropy as *AC+VE* and *AC*. We denote Distributional SAC (DSAC) [34] with/without vanilla entropy as *AC+UE+VE* and *AC+UE*. *AC+UE* is also denoted as *DAC*. The implementation details can be found in Appendix K.

**Experimental Details.** We demonstrate that the two types of regularized exploration in MaxEnt RL and CDRL play distinct roles in policy learning when employed simultaneously, including mutual improvement or potential interference. We perform our experiments for both DSAC (C51) in Figure 4 and DSAC (IQN) in Figure 7 of Appendix L.2, where the latter is used to examine the mutual impacts in quantile-based distributional RL heuristically.

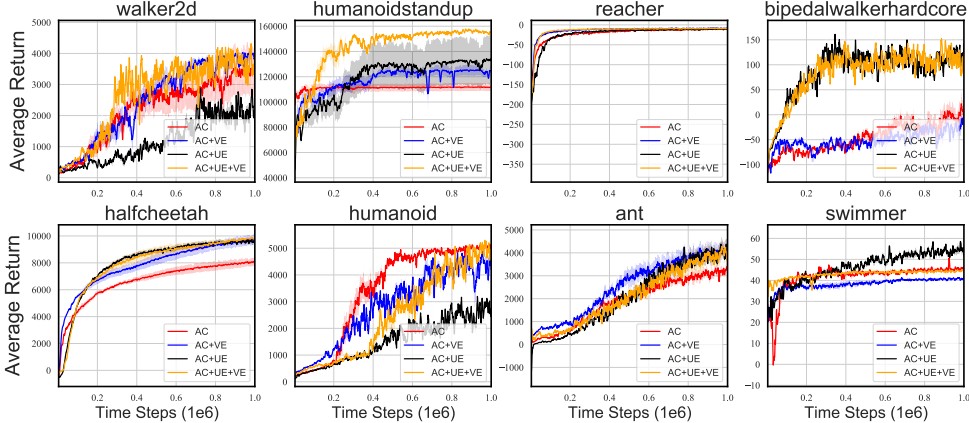

Figure 4: Learning curves of *AC*, *AC+VE* (SAC), *AC+UE* (DAC) and AC+UE+VE (DSAC) over five seeds across seven MuJoCo environments where the distributional RL part is based on C51. (**First Row**): Mutual Improvement. (**Second Row**): Potential Interference.

**Results.** In the first row of Figure 4, simultaneously employing uncertainty-aware and vanilla entropy regularization renders a mutual improvement. Conversely, the two kinds of regularizations, when adopted together, can also lead to performance degradation, as exhibited in the second row in Figure 4. For instance, AC+UE+VE outperforms both *AC+VE* (SAC) and *AC+UE* (DAC) on humanoidstandup, while suffering from performance degradation on Ant and Swimmer. We posit that the potential interference may result from distinct exploration directions in the policy learning for the two types of regularizations. SAC optimizes the policy to visit states with high entropy, while distributional RL updates the policy to explore states and the associated actions whose current return distribution estimate lags far behind the environment uncertainty in target returns.

## 7 Extension to Quantile Distributional Loss

As an extension, we consider decomposing the quantile distributional loss, which is also commonly used in distributional RL, such as QR-DQN and IQN. Due to space limitations, a more detailed description is deferred to Appendix M. The quantile distributional loss can be viewed as a variant of composite quantile loss [72]. We commit to decomposing it into a mean-related term and a residual

term, where the mean-related term is related to the expected quantile values. We demonstrate that minimizing the decomposed mean-related term is asymptotically mean-preserving [49] as the number of quantiles approaches infinity. The induced residual term, therefore, captures the information from the return distribution that excludes its expectation, serving as the benefit to explain the superiority of quantile-based distributional RL.

## 8 Conclusion and Discussion

In this study, we interpret the benefits of CDRL over classical RL as uncertainty-aware regularization via return density decomposition. In contrast to the exploration to encourage diverse actions in MaxEnt RL, the uncertainty-aware regularization in CDRL promotes exploring states where the environmental uncertainty is largely underestimated. Our study offers a novel exploration perspective to analyze the benefits of (categorical) distributional learning in RL.

**Limitation and Future Work.** The uncertainty-aware regularized exploration from distributional loss is mainly founded on CDRL. Although briefly examined in Section 7, it remains interesting yet challenging to extend our conclusion to general distributional RL, given that the analytical techniques, such as those in QR-DQN, are largely different from CDRL. We leave this extension for future work.

## Acknowledgements

This work was primarily conducted while Ke Sun was at the University of Alberta. Some revisions were completed during his postdoctoral research at Harvard University, where he was supported by NIH/NIDA P50DA054039, NIH/NIA 5P30AG073107-03 GY3 Pilots, and NIH/NIDCR UH3DE028723. Bei Jiang and Linglong Kong were partially supported by grants from the Canada CIFAR AI Chairs program, the Alberta Machine Intelligence Institute (AMII), and Natural Sciences and Engineering Council of Canada (NSERC), and Linglong Kong was also partially supported by grants from the Canada Research Chair program from NSERC. Xiaodong Yan was supported by the National Key R&D Program of China (No. 2023YFA1008701) and the National Natural Science Foundation of China (No. 12371292) equally. The authors express their gratitude for the insightful feedback provided by all reviewers and the area chairs, which significantly enhanced the initial version of this paper.

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

# Appendix

## Table of Contents

# A    Related Work: More Discussions about Uncertainty-oriented Exploration in RL

**Uncertainty in RL.** Uncertainty is ubiquitous in RL and sequential decision-making, and therefore harnessing uncertainty is always crucial in designing efficient algorithms [32]. In the literature of uncertainty quantification, uncertainty is often decomposed into two sources: *aleatoric uncertainty* and *epistemic uncertainty*.

- *Aleatoric uncertainty*, also called intrinsic or environmental uncertainty, originates from the stochastic or probabilistic nature of the environment, encompassing three main sources: stochastic transition dynamics, stochastic policy, and stochastic reward function. Aleatoric uncertainty is determined by the environment, which is thus irreducible. However, we can design more efficient algorithms by capturing more environmental uncertainty in the learning process, e.g., via distributional RL.

- *Epistemic uncertainty*, also called parametric uncertainty, often originates from the stochasticity in statistical estimation in the presence of limited data or incomplete knowledge. As opposed to aleatoric uncertainty, epistemic uncertainty is reducible and should decrease over more data, which contributes to a more reliable statistical estimation.

**Uncertainty-oriented Exploration.** There are a few survey papers that comprehensively summarize existing exploration approaches [29, 20]. Following [20], we classify the exploration strategies into two main categories: *uncertainty-oriented exploration* and *intrinsic motivation-oriented exploration*. The latter is inspired by psychology, which is not the focus of our study. Importantly, according to the two categories of uncertainty in RL, uncertainty-oriented exploration, which often applies *Optimism in the Face of Uncertainty* (OFU) principle, involves aleatoric and epistemic uncertainty.

- *Epistemic uncertainty-oriented exploration* takes advantage of the uncertainty in the (posterior) estimation of value functions. The typical exploration methods include Bayesian framework [45, 4, 38], Bootstrap [44], and Ensemble methods [30]. For instance, Bootstrapped DQN [44] maintains several independent Q-estimators and randomly samples one of them, enabling the agent to perform temporally extended exploration.

- *Aleatoric uncertainty-oriented exploration* aims to capture more environmental uncertainty from three sources of stochastic transition dynamics, stochastic policies, and stochastic reward function, all of which can be comprehensively integrated into return distribution. [9] employs Perturbed Quantile Regression (PQR) to promote the optimistic exploration within the distributional RL framework, while Decaying Left Truncated Variance (DLTV) [36] utilizes the variance from the learned return distributions. [58] investigates the approximate posterior sampling in distributional RL to encourage the exploration. By contrast, our primal goal in this study is to attribute the benefits of distributional RL to its intrinsic uncertainty-aware exploration we derived via return density decomposition instead of harnessing the learned return distribution to develop subsequent aleatoric uncertainty-oriented exploration strategies in [58, 36]. On the other hand, MaxEnt RL [16, 17, 18] utilizes the stochasticity of learned policy, one of the three sources in environmental uncertainty, to encourage diverse actions. Therefore, MaxEnt RL can also be categorized into the aleatoric uncertainty-oriented exploration, and it is thus intuitive and interesting to make a detailed comparison of the exploration effects between distributional RL and MaxEnt RL, conducted in Section 5.1 of our study.

# B    More Details about Categorical Distributional RL and Algorithm Description of C51

**Distributional Loss and Projection in CDRL.** Categorical Distributional RL [5] uses the heuristic projection operator $\Pi_{\mathcal{C}}$, which was defined as

$$\Pi_{\mathcal{C}}\left(\delta_y\right) = \begin{cases} \delta_{z_1} & y \leq z_1 \\ \frac{z_{i+1}-y}{z_{i+1}-z_i}\delta_{l_i} + \frac{y-z_i}{z_{i+1}-z_i}\delta_{z_{i+1}} & z_i < y \leq z_{i+1} \\ \delta_{z_N} & y > z_N \end{cases}, \tag{10}$$

After applying the distributional Bellman operator $\mathfrak{T}^{\pi}$ on the current return distribution $\eta^{\pi}(s,a)$ in each update, the resulting new distribution, which we denote as $\widetilde{\eta}^{\pi}(s,a)$, typically no longer lies

in the same (discrete) support with the original one on $\{z_i\}_{i=1}^N$. To maintain the same support, the underpinning of the KL divergence, CDRL additionally applies the projection operator $\Pi_{\mathcal{C}}$ on the new distribution $\widetilde{\eta}^\pi(s, a)$. This projection rule distributes the weight of $\delta_y$ across the original support points $\{z_i\}_{i=1}^N$ based on the linear interpolation. For example, if $y$ lies in between two support points $z_i$ and $z_{i+1}$, the probability mass on $y$ is split between $z_i$ and $z_{i+1}$ with the weight inversely proportional to its distance ratio to $z_i$ and $z_{i+1}$. Therefore, the projection extends affinely to finite mixtures of Dirac measures, such that for a mixture of Diracs $\sum_{i=1}^N p_i \delta_{y_i}$, we have $\Pi_{\mathcal{C}}\left(\sum_{i=1}^N p_i \delta_{y_i}\right) = \sum_{i=1}^N p_i \Pi_{\mathcal{C}}\left(\delta_{y_i}\right)$. The Cramér distance was recently studied as an alternative to the Wasserstein distances in the context of generative models [7]. Recall the definition of Cramér distance in the following.

**Definition 1.** *(Definition 3 [48]) The Cramér distance $\ell_2$ between two distributions $\nu_1, \nu_2 \in \mathscr{P}(\mathbb{R})$, with cumulative distribution functions $F_{\nu_1}, F_{\nu_2}$ respectively, is defined by:*

$$\ell_2\left(\nu_1, \nu_2\right) = \left(\int_{\mathbb{R}} \left(F_{\nu_1}(x) - F_{\nu_2}(x)\right)^2 \, \mathrm{d}x\right)^{1/2}.$$

*Further, the supremum-Cramér metric $\bar{\ell}_2$ is defined between two distribution functions $\eta, \mu \in \mathscr{P}(\mathbb{R})^{\mathcal{X} \times \mathcal{A}}$ by*

$$\bar{\ell}_2(\eta, \mu) = \sup_{(x,a) \in \mathcal{X} \times \mathcal{A}} \ell_2\left(\eta^{(x,a)}, \mu^{(x,a)}\right).$$

Thus, the contraction of categorical distributional RL can be guaranteed under Cramér distance:

**Proposition 4.** *(Proposition 2 [48]) The operator $\Pi_{\mathcal{C}}\mathcal{T}^\pi$ is a $\sqrt{\gamma}$-contraction in $\bar{\ell}_2$.*

An insight behind this conclusion is that Cramér distance endows a particular subset with a notion of orthogonal projection, and the orthogonal projection onto the subset is exactly the heuristic projection $\Pi_{\mathcal{C}}$ (Proposition 1 in [48]). [48] also states that the operator $\Pi_{\mathcal{C}}\mathcal{T}^\pi$ is contractive under Wasserstein distance.

**Description of CDRL Algorithm: C51.** With $N = 51$, C51 instantiates the CDRL algorithm. To elaborate the algorithm, we first introduce the pushforward measure $f_{\#}\nu \in \mathcal{P}(\mathbb{R})$ from Definition 1 in [48]. This pushforward measure shifts the support of the probability measure $\mu$ according to the map $f$, which is commonly used in distributional RL literature. In particular, we consider an affine shift map $f_{r,\gamma} : \mathbb{R} \to \mathbb{R}$, defined by $f_{r,\gamma}(x) = r + \gamma x$. As Algorithm 1 displays, we first apply the pushforward measure on the target return distribution $\widehat{\eta}(s', a^*)$ by affinely shifting its support points, leading to a new distribution $\widetilde{\eta}(s, a)$. Next, we project the support points of $\widetilde{\eta}(s, a)$ by employing $\Pi_{\mathcal{C}}$ onto the original support, allowing us to compute the KL divergence in the end. Notably, we decompose the distributional objective function on the KL loss $\mathrm{KL}(\widehat{\eta}_{\text{target}}(s, a) || \widehat{\eta}(s, a))$.

---

**Algorithm 1** CDRL Update (Adapted from Algorithm 1 in [48])

---

**Require**: Number of atoms $N$, e.g., $N = 51$ in C51, the categorical distribution $\widehat{\eta}(s, a) = \sum_{i=1}^N p_i^{s,a} \delta_{z_i}$ for the current return distribution.
**Input**: Sample transition $(s, a, r, s')$

 1: **if** Policy evaluation: **then**
 2:     $a^* \sim \pi(\cdot | s')$
 3: **else if** Control: **then**
 4:     $a^* \leftarrow \arg\max_{a' \in \mathcal{A}} \mathbb{E}_{R \sim \widehat{\eta}(s', a')}[R]$
 5: **end if**
 6: $\widetilde{\eta}(s, a) \leftarrow (f_{r,\gamma})_{\#}\widehat{\eta}(s', a^*)$ # Distributional Bellmen update by applying $\widehat{\mathfrak{T}}^\pi$
 7: $\widehat{\eta}_{\text{target}}(s, a) \leftarrow \Pi_{\mathcal{C}}\widetilde{\eta}(s, a)$ # Project target support points and then distribute the probabilities
**Output**: Compute the distributional loss $\mathrm{KL}(\widehat{\eta}_{\text{target}}(s, a) || \widehat{\eta}(s, a))$ # Choose KL divergence as $d_p$

---

## C   Explanation about the Efficacy of Neural FZI Framework

The main reason for adopting the Neural FZI framework in Section 4.1 to interpret the behavior of distributional RL is that it is by far the closest framework to the practical algorithms, to the best of

our knowledge. Neural FZI or FQI was initially proposed in the context of offline RL with a fixed dataset of transitions $(s, a, s', r)$, the leverage of the Neural FZI framework does not assume a static dataset and accommodates an evolving dataset when using an exploration mechanism. Specifically, our theoretical analysis focuses on the decomposition of the categorical distributional loss, but not on a formal convergence proof of Neural FZI under an evolving dataset. The related convergence proof might be complicated beyond classical assumptions on Neural FQI or FZI in an offline setting. This distinction is important: the goal of our analysis is to reveal how the categorical loss can be interpreted as a form of distributional entropy regularization in Proposition 2, regardless of whether the underlying data distribution changes over time. During each phase of Neural FZI, the return distribution can be updated based on newly collected transitions, allowing the exploration of new samples based on the decomposed distribution-matching regularization term

## D Proof of Proposition 1

**Proposition 1.**(Decomposition Validity) Denote $\widehat{p}^{s,a}(x \in \Delta_E) = p_E/\Delta$, where $p_E$ is the coefficient on the bin $\Delta_E$. $\widehat{\mu}^{s,a}(x) = \sum_{i=1}^{N} p_i^{\mu} \mathbb{1}(x \in \Delta_i)/\Delta$ is a valid density if and only if $\epsilon \geq 1 - p_E$.

*Proof.* Recap a valid probability density function requires non-negative and one-bounded probability in each bin and all probabilities should sum to 1. We start to prove all probabilities should sum to 1, which is straightforward by taking the integral of both sides of Eq 3:

$$
\int \widehat{p}^{s,a}(x)dx = (1 - \epsilon) \int \frac{\mathbb{1}(x \in \Delta_E)}{\Delta} dx + \epsilon \int \widehat{\mu}^{s,a}(x)dx
$$
$$
1 = (1 - \epsilon) + \epsilon \int \widehat{\mu}^{s,a}(x)dx, \tag{11}
$$

which directly implies $\int \widehat{\mu}^{s,a}(x)dx = 1$. Next, we show necessity and sufficiency of non-negative and one-bounded probability in each bin.

**Necessity.** (1) When $x \in \Delta_E$, Eq. 3 can simplified as $p_E/\Delta = (1 - \epsilon)/\Delta + \epsilon p_E^{\mu}/\Delta$, where $p_E^{\mu} = \widehat{\mu}(x \in \Delta_E)$. Thus, $p_E^{\mu} = \frac{p_E}{\epsilon} - \frac{1-\epsilon}{\epsilon} \geq 0$ if $\epsilon \geq 1 - p_E$. Obviously,

$$
p_E^{\mu} = \frac{p_E}{\epsilon} - \frac{1 - \epsilon}{\epsilon} \leq \frac{1}{\epsilon} - \frac{1 - \epsilon}{\epsilon} = 1, \tag{12}
$$

which is guaranteed by the validity of $\widehat{p}_E^{s,a}$. (2) When $x \notin \Delta_E$, we have $p_i/\Delta = \epsilon p_i^{\mu}/\Delta$, i.e.,When $x \notin \Delta_E$, We immediately have $p_i^{\mu} = \frac{p_i}{\epsilon} \leq \frac{1-p_E}{\epsilon} \leq 1$ when $\epsilon \geq 1 - p_E$. Also, $p_i^{\mu} = \frac{p_i}{\epsilon} \geq 0$.

**Sufficiency.** (1) When $x \in \Delta_E$, let $p_E^{\mu} = \frac{p_E}{\epsilon} - \frac{1-\epsilon}{\epsilon} \geq 0$, we have $\epsilon \geq 1 - p_E$. $p_E^{\mu} = \frac{p_E}{\epsilon} - \frac{1-\epsilon}{\epsilon} \leq 1$ in nature. (2) When $x \notin \Delta_E$, $p_i^{\mu} = \frac{p_i}{\epsilon} \geq 0$ in nature. Let $p_i^{\mu} = \frac{p_i}{\epsilon} \leq 1$, we have $p_i \leq \epsilon$. We need to take the intersection set of (1) and (2), and we find that

$$
\epsilon \geq 1 - p_E \Rightarrow \epsilon \geq 1 - p_E \geq p_i, \tag{13}
$$

which satisfies the condition in (2). Thus, the intersection set of (1) and (2) would be $\epsilon \geq 1 - p_E$.

In summary, as $\epsilon \geq 1 - p_E$ is both the necessary and sufficient condition, we have the conclusion that $\widehat{\mu}(x)$ is a valid probability density function $\iff \epsilon \geq 1 - p_E$.

$\square$

## E Equivalence between Categorical Parameterization and Histogram Density Estimation in Distributional RL

**Proposition 5.** *Suppose the target categorical distribution $c = \sum_{i=1}^{N} p_i \delta_{z_i}$ and the target histogram function $h(x) = \sum_{i=1}^{N} p_i \mathbb{1}(x \in \Delta_i)/\Delta$, updating the parameterized categorical distribution $c_\theta$ under KL divergence is equivalent to updating the parameterized histogram function $h_\theta$.*

*Proof.* For the histogram density estimator $h_\theta$ and the true target density function $p(x)$, we can simplify the KL divergence as follows.

$$
\begin{aligned}
D_{\mathrm{KL}}(h, h_\theta) &= \sum_{i=1}^{N} \int_{l_{i-1}}^{l_i} \frac{p_i(x)}{\Delta} \log \frac{\frac{p_i(x)}{\Delta}}{\frac{h_\theta^i}{\Delta}} dx \\
&= \sum_{i=1}^{N} \int_{l_{i-1}}^{l_i} \frac{p_i(x)}{\Delta} \log \frac{p_i(x)}{\Delta} dx - \sum_{i=1}^{N} \int_{l_{i-1}}^{l_i} \frac{p_i(x)}{\Delta} \log \frac{h_\theta^i}{\Delta} dx \qquad (14)\\
&\overset{(a)}{\propto} -\sum_{i=1}^{N} \int_{l_{i-1}}^{l_i} \frac{p_i(x)}{\Delta} \log \frac{h_\theta^i}{\Delta} dx \overset{(b)}{=} -\sum_{i=1}^{N} p_i \log \frac{h_\theta^i}{\Delta} \overset{(c)}{\propto} -\sum_{i=1}^{N} p_i \log h_\theta^i,
\end{aligned}
$$

where $h_\theta^i$ is determined by $i$ and $\theta$, which is independent of $x$. $(a)$ is true because the target distribution with all $p_i$ is fixed. $(b)$ follows because $p_i(x)$ remains constant for $x \in [l_i, l_{i+1}]$. Finally, $(c)$ holds as the remaining term involving $p_i$, and $\Delta$ is also constant.

On the other hand, we consider the KL-based objective function in learning categorical distribution estimator. Given the target categorical distribution $c = \sum_{i=1}^{N} p_i \delta_{z_i}$, where the probability $p_i$ is fixed for each atom $z_i$, we aim at updating the current categorical estimator $c_\theta$. Then, we have:

$$
D_{\mathrm{KL}}(c, c_\theta) = \sum_{i=1}^{N} p_i \log \frac{p_i}{c_\theta^i} = \sum_{i=1}^{N} p_i \log p_i - \sum_{i=1}^{N} p_i \log c_\theta^i \propto -\sum_{i=1}^{N} p_i \log c_\theta^i, \qquad (15)
$$

where $c_\theta = \sum_{i=1}^{N} c_\theta^i \delta_{z_i}$ is the current categorical estimator and $c_\theta^i$ is the learnable probability. By comparing the final loss function forms in Eq. 14 and Eq. 15, it turns out that they are equivalent as both $c_\theta^i$ and $h_\theta^i$ are the learnable probabilities, which are parameterized by the same neural network.

$\square$

**Remark.** In CDRL, we use a discrete categorical distribution with probabilities centered on the fixed atoms $\{z_i\}_{i=1}^{N}$. In contrast, the histogram density estimator in our analysis is a continuous function defined on $[z_0, z_N]$, enabling more nuanced analysis within continuous functions. Proposition 5 indicates that minimizing the KL divergence with the categorical distribution in Eq. 15 amounts to the cross-entropy loss with the parameterized histogram function in Eq. 14.

# F  Convergence Guarantee of Histogram Density Estimator in Distributional RL

**Histogram Function Parameterization Error: Uniform Convergence in Probability.** The previous discrete categorical parameterization error bound in [48] (Proposition 3) is derived between the true return distribution and the limiting return distribution denoted as $\eta_C$ iteratively updated via the Bellman operator $\Pi_C \mathfrak{T}^\pi$ *in expectation*, without considering an asymptotic analysis when the number of sampled $\{s_i, a_i\}_{i=1}^{n}$ pairs goes to infinity. As a complementary result, we provide a uniform convergence rate for the histogram density estimator in the context of distributional RL. In this particular analysis within this subsection, we denote $\widehat{p}_C^{s,a}$ as the density function estimator for the true limiting return distribution $\eta_C$ via $\Pi_C \mathfrak{T}^\pi$ with its true density $p_C^{s,a}$. In Theorem 2, we show that the sample-based histogram estimator $\widehat{p}_C^{s,a}$ can approximate any arbitrary continuous limiting density function $p_C^{s,a}$ under a mild condition. ***This ensures the use of a histogram density estimator in the implementation of our subsequent algorithm adapted from CDRL.***

**Theorem 2.** *(Uniform Convergence Rate in Probability) Suppose $p_C^{s,a}(x)$ is Lipschitz continuous, and the support of a random variable is partitioned by N bins with bin size $\Delta$. Then*

$$
\sup_x |\widehat{p}_C^{s,a}(x) - p_C^{s,a}(x)| = O(\Delta) + O_P\left(\sqrt{\frac{\log N}{n\Delta^2}}\right). \qquad (16)
$$

*Proof.* Our proof is mainly based on the non-parametric statistics analysis [61]. In particular, the difference of $\widehat{p}_C^{s,a}(x) - p_C^{s,a}(x)$ can be written as

$$
\widehat{p}_C^{s,a}(x) - p_C^{s,a}(x) = \underbrace{\mathbb{E}\left(\widehat{p}_C^{s,a}(x)\right) - p_C^{s,a}(x)}_{\text{bias}} + \underbrace{\widehat{p}_C^{s,a}(x) - \mathbb{E}\left(\widehat{p}_C^{s,a}(x)\right)}_{\text{stochastic variation}}. \qquad (17)
$$

**(1) The first bias term.** Without loss of generality, we consider $x \in \Delta_k$, we have

$$
\begin{aligned}
\mathbb{E}\left(\widehat{p}_{\mathcal{C}}^{s,a}(x)\right) &= \frac{P(X \in \Delta_k)}{\Delta} \\
&= \frac{\int_{l_0+(k-1)\Delta}^{l_0+k\Delta} p(y)dy}{\Delta} \\
&= \frac{F(l_0+(k-1)\Delta) - F(l_0+(k-1)\Delta)}{l_0+k\Delta - (l_0+(k-1)\Delta)} \\
&= p_{\mathcal{C}}^{s,a}(x'),
\end{aligned}
\tag{18}
$$

where the last equality is based on the mean value theorem. According to the L-Lipschitz continuity property, we have

$$
|\mathbb{E}\left(\widehat{p}_{\mathcal{C}}^{s,a}(x)\right) - p_{\mathcal{C}}^{s,a}(x)| = |p_{\mathcal{C}}^{s,a}(x') - p_{\mathcal{C}}^{s,a}(x)| \leq L|x' - x| \leq L\Delta
\tag{19}
$$

**(2) The second stochastic variation term.** If we let $x \in \Delta_k$, then $\widehat{p}_{\mathcal{C}}^{s,a} = p_k = \frac{1}{n}\sum_{i=1}^{n} \mathbb{1}(X_i \in \Delta_k)$, we thus have

$$
\begin{aligned}
&P\left(\sup_x |\widehat{p}_{\mathcal{C}}^{s,a}(x) - \mathbb{E}\left(\widehat{p}_{\mathcal{C}}^{s,a}(x)\right)| > \epsilon\right) \\
&= P\left(\max_{j=1,\cdots,N} \left|\frac{1}{n}\sum_{i=1}^{n} \mathbb{1}(X_i \in \Delta_j)/\Delta - P(X_i \in \Delta_j)/\Delta\right| > \epsilon\right) \\
&= P\left(\max_{j=1,\cdots,N} \left|\frac{1}{n}\sum_{i=1}^{n} \mathbb{1}(X_i \in \Delta_j) - P(X_i \in \Delta_j)\right| > \Delta\epsilon\right) \\
&\leq \sum_{j=1}^{N} P\left(\left|\frac{1}{n}\sum_{i=1}^{n} \mathbb{1}(X_i \in \Delta_j) - P(X_i \in \Delta_j)\right| > \Delta\epsilon\right) \\
&\leq N \cdot \exp\left(-2n\Delta^2\epsilon^2\right) \quad \text{(by Hoeffding's inequality)},
\end{aligned}
\tag{20}
$$

where in the last inequality we know that the indicator function is bounded in [0, 1]. We then let the last term be a constant independent of $N, n, \Delta$ and simplify the order of $\epsilon$. Then, we have:

$$
\sup_x |\widehat{p}_{\mathcal{C}}^{s,a}(x) - \mathbb{E}\left(\widehat{p}_{\mathcal{C}}^{s,a}(x)\right)| = O_P\left(\sqrt{\frac{\log N}{n\Delta^2}}\right)
\tag{21}
$$

In summary, as the above inequality holds for each $x$, we thus have the uniform convergence rate of a histogram density estimator

$$
\begin{aligned}
\sup_x |\widehat{p}_{\mathcal{C}}^{s,a}(x) - p_{\mathcal{C}}^{s,a}(x)| &\leq \sup_x |\mathbb{E}\left(\widehat{p}_{\mathcal{C}}^{s,a}(x)\right) - p_{\mathcal{C}}^{s,a}(x)| + \sup_x |\widehat{p}_{\mathcal{C}}^{s,a}(x) - \mathbb{E}\left(\widehat{p}_{\mathcal{C}}^{s,a}(x)\right)| \\
&= O(\Delta) + O_P\left(\sqrt{\frac{\log N}{n\Delta^2}}\right).
\end{aligned}
\tag{22}
$$

$\square$

# G   Discussion about KL Divergence in Distributional RL

## G.1   Properties of KL divergence in Distributional RL

**Remark on KL Divergence.** As stated in Section 3 of CDRL [5], when the categorical parameterization is applied after the projection operator $\Pi_{\mathcal{C}}$, the distributional Bellman operator $\mathfrak{T}^{\pi}$ has the contraction guarantee under Cramér distance or Wasserstein distance [48], albeit the direct use of a non-expansive KL divergence [41]. Similarly, our histogram density parameterization with the projection $\Pi_{\mathcal{C}}$ and KL divergence also enjoys a contraction property due to the equivalence between optimizing histogram function and categorical distribution analyzed in Appendix E. We summarize some properties of KL divergence in distributional RL in Proposition 6.

**Proposition 6.** *Given two probability measures $\mu$ and $\nu$, we define the supreme $D_{KL}$ as a functional $\mathcal{P}(\mathcal{X})^{\mathcal{S}\times\mathcal{A}} \times \mathcal{P}(\mathcal{X})^{\mathcal{S}\times\mathcal{A}} \to \mathbb{R}$, i.e., $D_{KL}^{\infty}(\mu,\nu) = \sup_{(s,a)\in\mathcal{S}\times\mathcal{A}} D_{KL}(\mu(s,a),\nu(s,a))$. we have:*

*(1) $\mathfrak{T}^{\pi}$ is a non-expansive distributional Bellman operator under $D_{KL}^{\infty}$, i.e.,*

$$D_{KL}^{\infty}(\mathfrak{T}^{\pi}Z_1,\mathfrak{T}^{\pi}Z_2) \leq D_{KL}^{\infty}(Z_1,Z_2), \tag{23}$$

*(2) $D_{KL}^{\infty}(Z_n,Z) \to 0$ implies the Wasserstein distance $W_p(Z_n,Z) \to 0$.*

*Proof.* We first assume $Z_{\theta}$ is absolutely continuous and the supports of two distributions in KL divergence have a negligible intersection [2], under which the KL divergence is well-defined.

(1) The contraction analysis of distributional Bellman operator $\mathfrak{T}^{\pi}$ under a distribution divergence $d_p$ depends on its *scale sensitive* (**S**) and *sum invariant* (**I**) properties [7, 5]. We say $d_p$ is scale sensitive (of order $\tau$) if there exists a $\tau > 0$, such that for all random variables $X, Y$ and a real value $a > 0$, $d_p(aX, aY) \leq |a|^{\tau} d_p(X, Y)$. $d_p$ has the sum invariant property if whenever a random variable $A$ is independent from $X, Y$, we have $d_p(A + X, A + Y) \leq d_p(X, Y)$. We first prove that the $D_{KL}$ is sum-invariant, which is based on the dual form of KL divergence via the variational representation [12, 1]:

$$D_{\text{KL}}(X,Y) = \sup_{f\in\mathcal{L}^b} \{\mathbb{E}_X[f(x)] - \log\left(\mathbb{E}_Y\left[e^{f(y)}\right]\right)\}, \tag{24}$$

where $\mathcal{L}^b$ is the space of bounded measurable functions. Consequently, we have

$$
\begin{aligned}
D_{\text{KL}}(A + X, A + Y) &= \sup_{f\in\mathcal{L}^b} \{\mathbb{E}_{Z_1=A+X}[f(z_1)] - \log\left(\mathbb{E}_{Z_2=A+Y}\left[e^{f(z_2)}\right]\right)\} \\
&\overset{(a)}{=} \sup_{f\in\mathcal{L}^b} \{\mathbb{E}_A\left[\mathbb{E}_X\left[f(x+a)\right]\right] - \log\left(\mathbb{E}_A\left[\mathbb{E}_Y\left[e^{f(y+a)}\right]\right]\right)\} \\
&\overset{(b)}{\leq} \sup_{f\in\mathcal{L}^b} \{\mathbb{E}_A\mathbb{E}_X[f(x+a)] - \mathbb{E}_A\log\left(\mathbb{E}_Y\left[e^{f(y+a)}\right]\right)\} \\
&= \sup_{f\in\mathcal{L}^b} \{\mathbb{E}_A[\mathbb{E}_X[f(x+a)] - \log\left(\mathbb{E}_Y\left[e^{f(y+a)}\right]\right)]\} \\
&\overset{(c)}{\leq} \mathbb{E}_A \sup_{f\in\mathcal{L}^b} \{\mathbb{E}_X[f(x+a)] - \log\left(\mathbb{E}_Y\left[e^{f(y+a)}\right]\right)\} \\
&\overset{(d)}{=} \mathbb{E}_A \sup_{g\in\mathcal{L}^b} \{\mathbb{E}_X[g(x)] - \log\left(\mathbb{E}_Y\left[e^{g(y)}\right]\right)\} \\
&= D_{\text{KL}}(X,Y),
\end{aligned}
\tag{25}
$$

where (a) results from the independence between $A$ and $X$ ($Y$). (b) and (c) rely on the Jensen inequality for the function $-\log$ and the operator $\sup$. (d) is because the translation is still within the same bounded functional space. Next, we show that $D_{\text{KL}}$ is not scale-sensitive, where we denote the probability density function of $X$ and $Y$ as $p$ and $q$.

$$D_{\text{KL}}(aX, aY) = \int_{-\infty}^{\infty} \frac{1}{a}p\left(\frac{x}{a}\right)\log\frac{\frac{1}{a}p\left(\frac{x}{a}\right)}{\frac{1}{a}q\left(\frac{x}{a}\right)}\mathrm{d}x = \int_{-\infty}^{\infty} p(y)\log\frac{p(y)}{q(y)}\mathrm{d}y = D_{\text{KL}}(X,Y) \tag{26}$$

Putting the two properties together and given two return distributions $Z_1(s,a)$ and $Z_2(s,a)$, we have the non-expansive contraction property of the supremal form of $D_{\text{KL}}$ as follows.

$$
\begin{aligned}
D_{\text{KL}}^{\infty}(\mathfrak{T}^{\pi}Z_1,\mathfrak{T}^{\pi}Z_2) &= \sup_{s,a} D_{\text{KL}}(\mathfrak{T}^{\pi}Z_1(s,a),\mathfrak{T}^{\pi}Z_2(s,a)) \\
&= \sup_{s,a} D_{\text{KL}}(R(s,a) + \gamma Z_1(s',a'), R(s,a) + \gamma Z_2(s',a')) \\
&\overset{(e)}{\leq} D_{\text{KL}}(\gamma Z_1(s',a'),\gamma Z_2(s',a')) \\
&\overset{(f)}{=} D_{\text{KL}}(Z_1(s',a'), Z_2(s',a')) \\
&\leq \sup_{s,a} D_{\text{KL}}(Z_1(s',a'), Z_2(s',a')) \\
&= D_{\text{KL}}^{\infty}(Z_1,Z_2),
\end{aligned}
\tag{27}
$$

where $(e)$ relies on the sum invariant property of $D_{\mathrm{KL}}$ and $(f)$ holds due to the non-scale sensitive property of $D_{\mathrm{KL}}$. By applying the well-known Banach fixed point theorem, we have a unique return distribution when convergence of distributional dynamic programming under $D_{\mathrm{KL}}^\infty$.

(2) By the definition of $D_{\mathrm{KL}}^\infty$, we have $\sup_{s,a} D_{\mathrm{KL}}(Z_n(s,a), Z(s,a)) \to 0$ implies $D_{\mathrm{KL}}(Z_n, Z) \to 0$. $D_{\mathrm{KL}}(Z_n, Z) \to 0$ implies the total variation distance $\delta(Z_n, Z) \to 0$ according to a straightforward application of Pinsker's inequality

$$\delta\left(Z_n, Z\right) \le \sqrt{\frac{1}{2} D_{\mathrm{KL}}\left(Z_n, Z\right)} \to 0, \quad \delta\left(Z, Z_n\right) \le \sqrt{\frac{1}{2} D_{\mathrm{KL}}\left(Z, Z_n\right)} \to 0 \tag{28}$$

Based on Theorem 2 in WGAN [3], $\delta(Z_n, Z) \to 0$ implies $W_p(Z_n, Z) \to 0$. This is trivial by recalling the fact that $\delta$ and $W$ give the strong and weak topologies on the dual of $(C(\mathcal{X}), \|\cdot\|_\infty)$ when restricted to $\mathrm{Prob}(\mathcal{X})$.

$\square$

## G.2 Equivalence between Cross-Entropy Loss and KL Divergence in Neural FZI

If the target density function in evaluating the KL divergence is not fixed, using cross-entropy loss instead of the KL divergence may underestimate the uncertainty of return since this simplification may fail to capture the exact shape or uncertainty spread of the true target return distribution. However, this underestimation issue does occur in our analysis. Particularly, the leverage of the target network in Neural FZI, which is fixed in the updating of each phase, guarantees that the KL divergence is *exactly* proportional to the cross-entropy loss. Figure 5 suggests that C51 with cross-entropy loss (DSAC_CE) behaves similarly to the vanilla C51 equipped with KL divergence (DSAC) in both three Atari games and MuJoCo environments with continuous action space.

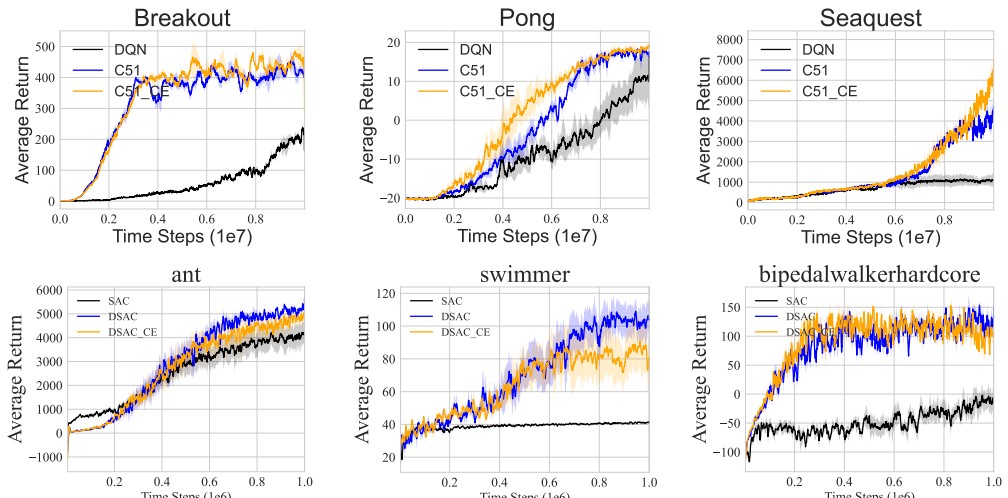

Figure 5: (**First row**) Learning curves of C51 under cross-entropy loss on Atari games over 3 seeds. (**Second row**) Learning curves of DSAC with C51 under cross-entropy loss on MuJoCo environments over five seeds.

## H Proof of Proposition 2

**Proposition 2** (Decomposed Neural FZI) Denote $q_\theta^{s,a}$ as the histogram density estimator of $Z_\theta^k(s,a)$ in Neural FZI. Based on the decomposition in Eq. 3 and the KL divergence as $d_p$, the Neural FZI process in Eq. 2 is simplified as

$$Z_\theta^{k+1} = \operatorname*{argmin}_{q_\theta} \frac{1}{n} \sum_{i=1}^n [\underbrace{-\log q_\theta^{s_i,a_i}(\Delta_E^i)}_{\text{Mean-Related Term}} + \underbrace{\alpha \mathcal{H}(\widehat{\mu}^{s_i', \pi_Z(s_i')}, q_\theta^{s_i,a_i})}_{\text{Regularization Term}}], \tag{29}$$

where $\alpha = \varepsilon/(1-\varepsilon) > 0$ and the mean-related term is negative log-likelihood centered on $\Delta_E^i$.

*Proof.* Firstly, given a fixed $p(x)$ we know that minimizing $D_{\mathrm{KL}}(p, q_\theta)$ is equivalent to minimizing $\mathcal{H}(p, q)$ by following

$$
\begin{aligned}
D_{\mathrm{KL}}(p, q_\theta) &= \sum_{i=1}^N \int_{l_{i-1}}^{l_i} \frac{p_i(x)}{\Delta} \log \frac{p^i(x)/\Delta}{q_\theta^i/\Delta}\, \mathrm{d}x \\
&= -\sum_{i=1}^N \int_{l_{i-1}}^{l_i} \frac{p_i(x)}{\Delta} \log \frac{q_\theta^i}{\Delta}\, \mathrm{d}x - \left( \sum_{i=1}^N \int_{l_{i-1}}^{l_i} \frac{p_i(x)}{\Delta} \log \frac{p^i(x)}{\Delta}\, \mathrm{d}x \right) \quad (30)\\
&= \mathcal{H}(p, q_\theta) - \mathcal{H}(p) \\
&\propto \mathcal{H}(p, q_\theta)
\end{aligned}
$$

where $p = \sum_{i=1}^N p_i(x) \mathbb{1}(x \in \Delta^i)/\Delta$ and $q_\theta = \sum_{i=1}^N q_i/\Delta$. Based on $\mathcal{H}(p, q_\theta)$, we use $p^{s_i', \pi_Z(s_i')}(x)$ to denote the target probability density function of the random variable $\mathcal{R}(s_i, a_i) + \gamma Z_{\theta^*}^k (s_i', \pi_Z(s_i'))$. Then, we can derive the objective function within each Neural FZI as

$$
\frac{1}{n} \sum_{i=1}^n \mathcal{H}(p^{s_i', \pi_Z(s_i')}, q_\theta^{s_i, a_i})
$$

$$
= \frac{1}{n} \sum_{i=1}^n \left( -(1-\epsilon) \sum_{j=1}^N \int_{l_{j-1}}^{l_j} \frac{\mathbb{1}(x \in \Delta_E^i)}{\Delta} \log \frac{q_\theta^{s_i, a_i}(\Delta_j)}{\Delta} dx - \epsilon \sum_{j=1}^N \int_{l_{j-1}}^{l_j} \frac{p_j^\mu}{\Delta} \log \frac{q_\theta^{s_i, a_i}(\Delta_j)}{\Delta} dx \right)
$$

$$
= \frac{1}{n} \sum_{i=1}^n \left( (1-\epsilon)(-\log q_\theta^{s_i, a_i}(\Delta_E^i)) + \epsilon \mathcal{H}(\widehat{\mu}^{s_i', \pi_Z(s_i')}, q_\theta^{s_i, a_i}) \right) + (1-\epsilon)\Delta
$$

$$
\propto \frac{1}{n} \sum_{i=1}^n \left( -\log q_\theta^{s_i, a_i}(\Delta_E^i) + \alpha \mathcal{H}(\widehat{\mu}^{s_i', \pi_Z(s_i')}, q_\theta^{s_i, a_i}) \right), \text{ where } \alpha = \frac{\epsilon}{1-\epsilon} > 0
$$

(31)

where recall that $\widehat{\mu}^{s_i', \pi_Z(s_i')} = \sum_{i=1}^N p_i^\mu(x) \mathbb{1}(x \in \Delta_i)/\Delta = \sum_{i=1}^N p_i^\mu/\Delta$ for conciseness and denote $q_\theta^{s_i, a_i} = \sum_{j=1}^N q_\theta^{s_i, a_i}(\Delta_j)/\Delta$. The cross-entropy $\mathcal{H}(\widehat{\mu}^{s_i', \pi_Z(s_i')}, q_\theta^{s_i, a_i})$ is based on the discrete distribution when $i = 1, ..., N$. $\Delta_E^i$ represent the interval that $\mathbb{E}\left[ \mathcal{R}(s_i, a_i) + \gamma Z_{\theta^*}^k (s_i', \pi_Z(s_i')) \right]$ falls into, i.e., $\mathbb{E}\left[ \mathcal{R}(s_i, a_i) + \gamma Z_{\theta^*}^k (s_i', \pi_Z(s_i')) \right] \in \Delta_E^i$. $\qquad \square$

## I  Proof of Proposition 3

**Proposition 3** (Equivalence between **the Mean-Related term** in Decomposed Neural FZI and Neural FQI) In Eq. 29, assume the function class $\{Z_\theta : \theta \in \Theta\}$ is sufficiently large such that it contains the target $\{Y_i^k\}_{i=1}^n$, when $\Delta \to 0$, for all $k$, minimizing **the mean-related term** in Eq. 29 implies

$$
P(Z_\theta^{k+1}(s, a) = \mathcal{T}^{\mathrm{opt}} Q_{\theta^*}^k(s, a)) = 1, \quad \text{and} \quad \int_{-\infty}^{+\infty} \left| F_{q_\theta}(x) - F_{\delta_{\mathcal{T}^{\mathrm{opt}} Q_{\theta^*}^k(s, a)}}(x) \right| dx = o(\Delta),
$$

(32)

where $\mathcal{T}^{\mathrm{opt}} Q_{\theta^*}^k(s, a)$ is the scalar-valued target in the k-th phase of Neural FQI, and $\delta_{\mathcal{T}^{\mathrm{opt}} Q_{\theta^*}^k(s, a)}$ is the Dirac delta function defined on the scalar $\mathcal{T}^{\mathrm{opt}} Q_{\theta^*}^k(s, a)$.

*Proof.* **Limiting Case.** Firstly, we define the distributional Bellman optimality operator $\mathfrak{T}^{\mathrm{opt}}$ as follows:

$$
\mathfrak{T}^{\mathrm{opt}} Z(s, a) \overset{D}{=} \mathcal{R}(s, a) + \gamma Z(S', a^*), \tag{33}
$$

where $S' \sim P(\cdot \mid s, a)$ and $a^* = \underset{a'}{\mathrm{argmax}} \mathbb{E}[Z(S', a')]$. If $\{Z_\theta : \theta \in \Theta\}$ is sufficiently large enough such that it contains $\mathfrak{T}^{\mathrm{opt}} Z_{\theta^*} (\{Y_i^k\}_{i=1}^n)$, then optimizing Neural FZI in Eq. 2 leads to $Z_\theta^{k+1} = \mathfrak{T}^{\mathrm{opt}} Z_{\theta^*}$.

Secondly, we apply the return density decomposition on the target histogram function $\widehat{p}^{s, a}(x)$. Consider the parameterized histogram density function $h_\theta$ and denote $h_\theta^E/\Delta$ as the bin height in the

bin $\Delta_E$, under the KL divergence between the first histogram function $\mathbb{1}(x \in \Delta_E)$ with $h_\theta(x)$, the objective function is simplified as

$$D_{\text{KL}}(\mathbb{1}(x \in \Delta_E)/\Delta, h_\theta(x)) = -\int_{x \in \Delta_E} \frac{1}{\Delta} \log \frac{\frac{h_\theta^E}{\Delta}}{\frac{1}{\Delta}} dx \quad = -\log h_\theta^E \tag{34}$$

Since $\{Z_\theta : \theta \in \Theta\}$ is sufficiently large enough that can represent the pdf of $\{Y_i^k\}_{i=1}^n$, it also implies that $\{Z_\theta : \theta \in \Theta\}$ can represent the mean-related term part in its pdf via the return density decomposition. The KL minimizer would be $\widehat{h}_\theta = \mathbb{1}(x \in \Delta_E)/\Delta$ in expectation. Then, $\lim_{\Delta \to 0} \arg\min_{h_\theta} D_{\text{KL}}(\mathbb{1}(x \in \Delta_E)/\Delta, h_\theta(x)) = \delta_{\mathbb{E}[Z^{\text{target}}(s,a)]}$, where $\delta_{\mathbb{E}[Z^{\text{target}}(s,a)]}$ is a Dirac Delta function centered at $\mathbb{E}[Z^{\text{target}}(s,a)]$ and can be viewed as a generalized probability density function. That being said, the limiting probability density function (pdf) converges to a Dirac delta function at $\mathbb{E}[Z^{\text{target}}(s,a)]$. The limit behavior from a histogram function $\widehat{p}$ to a continuous one for $Z^{\text{target}}$ is guaranteed by Theorem 2, and this also applies from $h_\theta$ to $Z_\theta$. In Neural FZI, we have $Z^{\text{target}} = \mathfrak{T}^{\text{opt}} Z_{\theta^*}$. Here, we use $Z_\theta^{k+1}(s,a)$ as the random variable whose cdf is the limiting distribution. According to the definition of the Dirac function, in the limiting case where $\Delta \to 0$, we attain that

$$\mathbb{P}(Z_\theta^{k+1}(s,a) = \mathbb{E}\left[\mathfrak{T}^{\text{opt}} Z_{\theta^*}^k(s,a)\right]) = 1. \tag{35}$$

This is because the pdf of the limiting return random variable $Z_\theta^{k+1}(s,a)$ is a Dirac delta function, which implies that the random variable takes this constant value with probability one. Due to the linearity of expectation in Lemma 4 of [5], we have

$$\mathbb{E}\left[\mathfrak{T}^{\text{opt}} Z_{\theta^*}^k(s,a)\right] = \mathfrak{T}^{\text{opt}} \mathbb{E}\left[Z_{\theta^*}^k(s,a)\right] = \mathcal{T}^{\text{opt}} Q_{\theta^*}^k(s,a) \tag{36}$$

Finally, we obtain the convergence in probability one in the limiting case:

$$\mathbb{P}(Z_\theta^{k+1}(s,a) = \mathcal{T}^{\text{opt}} Q_{\theta^*}^k(s,a)) = 1 \quad \text{as } \Delta \to 0 \tag{37}$$

**Convergence in Distribution.** The connection established above is in the limiting case. Alternatively, we can provide more formal proof by using the language of convergence in distribution. Here, we use $Z_{\theta,\Delta}^{k+1}$ to replace $Z_\theta^{k+1}$ to explicitly consider its asymptotic behavior. According to the fact that $\infty\{x \in \Delta_E\}/\Delta$ is the optimizer when minimizing the mean-related term in Eq. 29 given a fixed $\Delta$, the convergence in distribution is:

$$\lim_{\Delta \to 0} \mathcal{D}(Z_{\theta,\Delta}^{k+1}) = \lim_{\Delta \to 0} \mathcal{D}(\mathbb{1}\{x \in \Delta_E\}/\Delta) = \mathcal{D}(\delta_{\mathcal{T}^{\text{opt}} Q_{\theta^*}^k(s,a)}), \tag{38}$$

where $\delta_{\mathcal{T}^{\text{opt}} Q_{\theta^*}^k(s,a)}$ is the Dirac Delta function centered at $\mathcal{T}^{\text{opt}} Q_{\theta^*}^k(s,a)$. $\mathcal{D}(\delta_{\mathcal{T}^{\text{opt}} Q_{\theta^*}^k(s,a)})$ is the corresponding step function, where $\mathcal{D}(\delta_{\mathcal{T}^{\text{opt}} Q_{\theta^*}^k(s,a)})(x) = 1$ if $x \geq \mathcal{T}^{\text{opt}} Q_{\theta^*}^k(s,a)$, and equals 0 otherwise. Note that the convergence in distribution in terms of the Dirac delta function implies that $\mathbb{P}(Z_\theta^{k+1}(s,a) = \mathcal{T}^{\text{opt}} Q_{\theta^*}^k(s,a)) = 1$ as $\Delta \to 0$ in Eq 37.

**Convergence Rate.** In order to characterize how the difference varies when $\Delta \to 0$, we further define $\Delta_E = [l_e, l_{e+1})$ and we have:

$$\int_{-\infty}^{+\infty} \left| F_{q_\theta}(x) - F_{\delta_{\mathcal{T}^{\text{opt}} Q_{\theta^*}^k(s,a)}}(x) \right| dx = \frac{1}{2\Delta} \left( \left(\mathcal{T}^{\text{opt}} Q_{\theta^*}^k(s,a) - l_e\right)^2 + \left(l_{e+1} - \mathcal{T}^{\text{opt}} Q_{\theta^*}^k(s,a)\right)^2 \right)$$
$$= \frac{1}{2\Delta}(a^2 + (\Delta - a)^2)$$
$$\leq \Delta/2$$
$$= o(\Delta), \tag{39}$$

where $\mathcal{T}^{\text{opt}} Q_{\theta^*}^k(s,a) = \mathbb{E}\left[\mathfrak{T}^{\text{opt}} Z_{\theta^*}^k(s,a)\right] \in \Delta_E$ and we denote $a = \mathcal{T}^{\text{opt}} Q_{\theta^*}^k(s,a) - l_e$. The first equality holds as $q_\theta(x)$, the KL minimizer while minimizing the mean-related term, will follow a uniform distribution on $\Delta_E$, i.e., $\widehat{q}_\theta = \mathbb{1}(x \in \Delta_E)/\Delta$. Thus, the integral of LHS would be the area of two centralized triangles accordingly. The inequality holds as the maximizer is obtained when $a = \Delta$ or 0. The result implies that the convergence rate in distribution difference is $o(\Delta)$.

$\square$

# J  Convergence Proof of DERPI in Theorem 1

## J.1  Proof of Distribution-Entropy-Regularized Policy Evaluation in Lemma 1

**Lemma 1**(Distribution-Entropy-Regularized Policy Evaluation) Consider the distribution-entropy-regularized Bellman operator $\mathcal{T}_d^\pi$ in Eq. 7 and assume $\mathcal{H}(\mu^{\mathbf{s}_t,\mathbf{a}_t}, q_\theta^{\mathbf{s}_t,\mathbf{a}_t})$ is bounded for all $(\mathbf{s}_t, \mathbf{a}_t) \in \mathcal{S} \times \mathcal{A}$. Define $Q^{k+1} = \mathcal{T}_d^\pi Q^k$, then $Q^{k+1}$ will converge to a *corrected* Q-value of $\pi$ as $k \to \infty$ with the new objective function $J'(\pi)$ defined as

$$J'(\pi) = \sum_{t=0}^{T} \mathbb{E}_{(\mathbf{s}_t,\mathbf{a}_t)\sim\rho_\pi} \left[ r(\mathbf{s}_t, \mathbf{a}_t) + \gamma f(\mathcal{H}(\mu^{\mathbf{s}_t,\mathbf{a}_t}, q_\theta^{\mathbf{s}_t,\mathbf{a}_t})) \right].$$

*Proof.* Firstly, we plug in $V(\mathbf{s}_{t+1})$ into RHS of the iteration in Eq. 7, then we obtain

$\mathcal{T}_d^\pi Q(\mathbf{s}_t, \mathbf{a}_t)$
$= r(\mathbf{s}_t, \mathbf{a}_t) + \gamma \mathbb{E}_{\mathbf{s}_{t+1}\sim P(\cdot|\mathbf{s}_t,\mathbf{a}_t)} [V(\mathbf{s}_{t+1})]$
$= r(\mathbf{s}_t, \mathbf{a}_t) + \gamma \mathbb{E}_{\mathbf{s}_{t+1}\sim P(\cdot|\mathbf{s}_t,\mathbf{a}_t),\mathbf{a}_{t+1}\sim\pi} \left[ f(\mathcal{H}(\mu^{\mathbf{s}_{t+1},\mathbf{a}_{t+1}}, q_\theta^{\mathbf{s}_{t+1},\mathbf{a}_{t+1}})) \right] + \gamma \mathbb{E}_{\mathbf{s}_{t+1}\sim P(\cdot|\mathbf{s}_t,\mathbf{a}_t),\mathbf{a}_{t+1}\sim\pi} [Q(\mathbf{s}_{t+1}, \mathbf{a}_{t+1})]$
$\triangleq r_\pi(\mathbf{s}_t, \mathbf{a}_t) + \gamma \mathbb{E}_{\mathbf{s}_{t+1}\sim P(\cdot|\mathbf{s}_t,\mathbf{a}_t),\mathbf{a}_{t+1}\sim\pi} [Q(\mathbf{s}_{t+1}, \mathbf{a}_{t+1})],$

$$\tag{40}$$

where $r_\pi(\mathbf{s}_t, \mathbf{a}_t) \triangleq r(\mathbf{s}_t, \mathbf{a}_t) + \gamma \mathbb{E}_{\mathbf{s}_{t+1}\sim P(\cdot|\mathbf{s}_t,\mathbf{a}_t),\mathbf{a}_{t+1}\sim\pi} \left[ f(\mathcal{H}(\mu^{\mathbf{s}_{t+1},\mathbf{a}_{t+1}}, q_\theta^{\mathbf{s}_{t+1},\mathbf{a}_{t+1}})) \right]$ is the entropy augmented reward. Applying the standard convergence results for policy evaluation [57], we can attain that this Bellman updating under $\mathcal{T}_d^\pi$ is convergent under the assumption of $|\mathcal{A}| < \infty$ and bounded entropy augmented rewards $r_\pi$. $\qquad\square$

## J.2  Policy Improvement with Proof

**Lemma 2.** *(Distribution-Entropy-Regularized Policy Improvement) Let $\pi \in \Pi$ and a new policy $\pi_{new}$ be updated via the policy improvement step in the policy optimization:* $\pi_{new} = \arg\max_{\pi'\in\Pi} \mathbb{E}_{\mathbf{a}_t\sim\pi'} [Q^{\pi_{old}}(\mathbf{s}_t, \mathbf{a}_t) + f(\mathcal{H}(\mu^{\mathbf{s}_t,\mathbf{a}_t}, q_\theta^{\mathbf{s}_t,\mathbf{a}_t}))]$. *Then $Q^{\pi_{new}}(\mathbf{s}_t, \mathbf{a}_t) \geq Q^{\pi_{old}}(\mathbf{s}_t, \mathbf{a}_t)$ for all $(\mathbf{s}_t, \mathbf{a}_t) \in \mathcal{S} \times \mathcal{A}$ with $|\mathcal{A}| < \infty$.*

*Proof.* The policy improvement in Lemma 2 implies that

$\mathbb{E}_{\mathbf{a}_t\sim\pi_{new}} [Q^{\pi_{old}}(\mathbf{s}_t, \mathbf{a}_t) + f(\mathcal{H}(\mu^{\mathbf{s}_t,\mathbf{a}_t}, q_\theta^{\mathbf{s}_t,\mathbf{a}_t}))] \geq \mathbb{E}_{\mathbf{a}_t\sim\pi_{old}} [Q^{\pi_{old}}(\mathbf{s}_t, \mathbf{a}_t) + f(\mathcal{H}(\mu^{\mathbf{s}_t,\mathbf{a}_t}, q_\theta^{\mathbf{s}_t,\mathbf{a}_t}))].$

We consider the Bellman equation via the distribution-entropy-regularized Bellman operator $\mathcal{T}_{sd}^\pi$:

$Q^{\pi_{old}}(\mathbf{s}_t, \mathbf{a}_t)$
$\triangleq r(\mathbf{s}_t, \mathbf{a}_t) + \gamma \mathbb{E}_{\mathbf{s}_{t+1}\sim P} [V^{\pi_{old}}(\mathbf{s}_{t+1})]$
$= r(\mathbf{s}_t, \mathbf{a}_t) + \gamma \mathbb{E}_{\mathbf{s}_{t+1}\sim P} \left[ \mathbb{E}_{\mathbf{a}_{t+1}\sim\pi_{old}} [f(\mathcal{H}(\mu^{\mathbf{s}_{t+1},\mathbf{a}_{t+1}}, q_\theta^{\mathbf{s}_{t+1},\mathbf{a}_{t+1}})) + Q^{\pi_{old}}(\mathbf{s}_{t+1}, \mathbf{a}_{t+1})] \right]$
$\leq r(\mathbf{s}_t, \mathbf{a}_t) + \gamma \mathbb{E}_{\mathbf{s}_{t+1}\sim P} \left[ \mathbb{E}_{\mathbf{a}_{t+1}\sim\pi_{new}} [f(\mathcal{H}(\mu^{\mathbf{s}_{t+1},\mathbf{a}_{t+1}}, q_\theta^{\mathbf{s}_{t+1},\mathbf{a}_{t+1}})) + Q^{\pi_{old}}(\mathbf{s}_{t+1}, \mathbf{a}_{t+1})] \right]$
$= r(\mathbf{s}_t, \mathbf{a}_t) + \gamma \mathbb{E}_{\mathbf{s}_{t+1}\sim P,\mathbf{a}_{t+1}\sim\pi_{new}} \left[ f(\mathcal{H}(\mu^{\mathbf{s}_{t+1},\mathbf{a}_{t+1}}, q_\theta^{\mathbf{s}_{t+1},\mathbf{a}_{t+1}})) \right] + \gamma \mathbb{E}_{\mathbf{s}_{t+1}\sim P,\mathbf{a}_{t+1}\sim\pi_{new}} [Q^{\pi_{old}}(\mathbf{s}_{t+1}, \mathbf{a}_{t+1})]$
$= r_{\pi_{new}}(\mathbf{s}_t, \mathbf{a}_t) + \gamma \mathbb{E}_{\mathbf{s}_{t+1}\sim P,\mathbf{a}_{t+1}\sim\pi_{new}} [Q^{\pi_{old}}(\mathbf{s}_{t+1}, \mathbf{a}_{t+1})]$
$\vdots$
$\leq Q^{\pi_{new}}(\mathbf{s}_{t+1}, \mathbf{a}_{t+1}),$

$$\tag{41}$$

where $Q^{\pi_{old}}(\mathbf{s}_{t+1}, \mathbf{a}_{t+1})$ indicates that the future actions are taking following $\pi_{old}$, given $\mathbf{s}_{s+t}$ and $\mathbf{a}_{t+1}$. We have repeated expanded $Q^{\pi_{old}}$ on the RHS by applying the distribution-entropy-regularized distributional Bellman operator. Each following step will then incorporate the actions following the new policy. Convergence to $Q^{\pi_{new}}$ follows from Lemma 1. $\qquad\square$

## J.3  Proof of DERPI in Theorem 1

**Theorem 1** (Distribution-Entropy-Regularized Policy Iteration) Repeatedly applying distribution-entropy-regularized policy evaluation in Eq. 7 and the policy improvement, the policy converges to an optimal policy $\pi^*$ such that $Q^{\pi^*}(\mathbf{s}_t, \mathbf{a}_t) \geq Q^\pi(\mathbf{s}_t, \mathbf{a}_t)$ for all $\pi \in \Pi$.

*Proof.* The proof is similar to soft policy iteration [17]. For completeness, we provide the proof here. By Lemma 2, as the number of iteration increases, the sequence $Q^{\pi_i}$ at $i$-th iteration is monotonically increasing. Since we assume the uncertainty-aware entropy is bounded, the $Q^\pi$ is thus bounded as the rewards are bounded. Hence, the sequence will converge to some $\pi^*$. Further, we prove that $\pi^*$ is in fact optimal. At the convergence point, for all $\pi \in \Pi$, it must be case that:

$$\mathbb{E}_{\mathbf{a}_t \sim \pi^*} [Q^{\pi_{\text{old}}} (\mathbf{s}_t, \mathbf{a}_t)] \geq \mathbb{E}_{\mathbf{a}_t \sim \pi} [Q^{\pi_{\text{old}}} (\mathbf{s}_t, \mathbf{a}_t)].$$

According to the proof in Lemma 2, we can attain $Q^{\pi^*}(\mathbf{s}_t, \mathbf{a}_t) > Q^\pi(\mathbf{s}_t, \mathbf{a}_t)$ for $(\mathbf{s}_t, \mathbf{a}_t)$. That is to say, the "corrected" value function of any other policy in $\Pi$ is lower than the converged policy, indicating that $\pi^*$ is optimal. □

### J.4  Discussion about DERPI with Varying $q_\theta$

In the tabular setting, we have shown that the convergence of DERPI holds given a fixed $q_\theta$. The primary goal for us to derive this convergence result is to demonstrate the uncertainty-aware regularized exploration promoted by the decomposed regularization from (categorical) distributional loss. If we hope to develop a further algorithm in the function approximation, we need to consider how to interplay a parameterized Q function, policy, and $q$. For example, we may leverage separate neural network works for each component. Alternatively, we can use one single neural network to represent the whole return distribution ($q_\theta$) and then take the expectation to evaluate the Q function.

## K  Implementation Details

### K.1  More Descriptions of Baselines Algorithms

**Algorithms in Section 6.1.**

- DQN [39] and C51 [5]
- $\mathcal{H}(\mu, q_\theta)(\varepsilon = X)$: a variant of C51 algorithm, where we replace the original target histogram function $\widehat{p}^{s,a}$ with the induced $\widehat{\mu}^{s,a}$ for each $(s, a)$ pair in the update. By varying $\varepsilon = X$, $\mathcal{H}(\mu, q_\theta)$ relies on the distributional loss to different extents in the RL learning. For examples, when $\varepsilon = 1$, $\mathcal{H}(\mu, q_\theta)(\varepsilon = X)$ degenerates to the vanilla C51 algorithm. On the contrary, decreasing $\varepsilon$ in $\mathcal{H}(\mu, q_\theta)$ will reduce the leverage of knowledge from the distributional loss, leading to performance degradation in a distributional learning context.

**Algorithms in Section 6.2.**

- AC: This implementation is the same as AC.
- AC+VE: This is exactly the standard SAC algorithm.
- AC+UE: This implementation is also the same as DAC (C51), where we use a distributional critic loss in the AC algorithm.
- AC+UE+VE: Based on the SAC algorithm, i.e., AC+VE, we additionally use the distribution objective in C51 as the critic loss.

### K.2  Replacing $\epsilon$ with the ratio $\varepsilon$ for Visualization

$\varepsilon$ shares the same utility as $\epsilon$, but it is more convenient in implementation. $\varepsilon$ is defined as the mass proportion centered at the bin that contains the expectation *when transporting the mass to other bins*. A large proportion probability $\varepsilon$, which transports less mass to other bins, corresponds to a large $\epsilon$ in Eq. 3. Increasing $\varepsilon$ indicates that the decomposed algorithm performs more similarly to a pure CDRL algorithm. As Proposition 1 elucidates, the return density decomposition requires that $\epsilon$ exceed certain thresholds to ensure the resultant decomposed $\widehat{\mu}^{s,a}$ qualifies as a valid density function. In practice, pinpointing this lower boundary for $\epsilon$ in each iteration to regulate its range could be prohibitively time-intensive. A more pragmatic approach involves redistributing the mass from the bin that contains the expectation to other bins in specified ratios, thereby introducing the corresponding ratio term $\varepsilon$. By varying $\varepsilon$ from 0 to 1, it invariably meets the validity condition outlined in Proposition 1, thereby streamlining the process for conducting ablation studies concerning $\widehat{\mu}^{s,a}$ as demonstrated in Figure 3.

To delineate the relationship between the ratio $\varepsilon$ and the coefficient $\epsilon$ in constructing $\widehat{\mu}^{s,a}$, after some calculations we establish their equivalence as follows:

$$\varepsilon = \frac{p_E - (1 - \epsilon)}{p_E \epsilon}, \tag{42}$$

where $p_E$ represents the weighting assigned to the bin $\Delta_E$ as specified in Proposition 1. The resulting $\varepsilon \in [0, 1]$ has a monotonically increasing relationship with $\epsilon$. In addition, $\epsilon = 1$ implies $\varepsilon = 1$. These properties facilitate the visualization without undermining our conclusion.

**Decomposition Details.** By varying $\varepsilon$, we can evaluate $\epsilon$ via the transformation equation in Eq. 42, which guarantees the validity of return density decomposition. Next, under different $\epsilon$, we compute the induced histogram density $\widehat{\mu}^{s,a}$ via the return density decomposition in Eq. 3:

$$\widehat{\mu}^{s,a}(x) = \widehat{p}^{s,a}(x \notin \Delta_E)/\epsilon + \widehat{p}^{s,a}(x \in \Delta_E)\varepsilon, \tag{43}$$

where combines Eq. 3 and Eq. 42. Importantly, by summing all the probabilities of $p_i^\mu$ in $\mu$, we have:

$$\sum_{i=1}^{n} p_i^\mu = \frac{1 - p_E}{\epsilon} + \frac{p_E - (1 - \epsilon)}{\epsilon} = 1. \tag{44}$$

This substantiates the validity of our decomposition by using $\varepsilon$ instead of $\epsilon$ for visualization. Next, we replace $\widehat{p}^{s,a}$ with $\widehat{\mu}^{s,a}$ in C51 or the critic loss in Distributional AC (C51) as the decomposed algorithm $\mathcal{H}(\mu, q_\theta)$ and compare the performance of all considered algorithms. Please refer to the code in the implementation for more details.

### K.3 Hyper-parameters and Network structure

Our implementation is adapted from the popular RLKit platform. For Distributional SAC with C51, we use 51 atoms similar to the C51 [5]. For distributional SAC with quantile regression, instead of using fixed quantiles in QR-DQN, we leverage the quantile fraction generation based on IQN [10] that uniformly samples quantile fractions in order to approximate the full quantile function. In particular, we fix the number of quantile fractions as $N$ and keep them in ascending order. Besides, we adapt

Table 1: Hyper-parameters Sheet.

| Hyperparameter | Value |
|---|---|
| *Shared* | |
| Policy network learning rate | 3e-4 |
| (Quantile) Value network learning rate | 3e-4 |
| Optimization | Adam |
| Discount factor | 0.99 |
| Target smoothing | 5e-3 |
| Batch size | 256 |
| Replay buffer size | 1e6 |
| Minimum steps before training | 1e4 |
| *DSAC with C51* | |
| Number of Atoms ($N$) | 51 |
| *DSAC with IQN* | |
| Number of quantile fractions ($N$) | 32 |
| Quantile fraction embedding size | 64 |
| Huber regression threshold | 1 |

| Hyperparameter | Temperature Parameter $\beta$ | Max episode lenght |
|---|---|---|
| Walker2d-v2 | 0.2 | 1000 |
| Swimmer-v2 | 0.2 | 1000 |
| Reacher-v2 | 0.2 | 1000 |
| Ant-v2 | 0.2 | 1000 |
| HalfCheetah-v2 | 0.2 | 1000 |
| Humanoid-v2 | 0.05 | 1000 |
| HumanoidStandup-v2 | 0.05 | 1000 |
| BipedalWalkerHardcore-v2 | 0.002 | 2000 |

the sampling as $\tau_0 = 0, \tau_i = e_i / \sum_{i=0}^{N-1} e_i$, where $\epsilon_i \in U[0,1], i = 1, ..., N$. We adopt the same hyper-parameters, which are listed in Table 1 and network structure as in the original distributional SAC paper [34].

## L   Experiments Results

### L.1   Uncertainty-aware Regularization Effect by Varying $\epsilon$ in Actor Critic

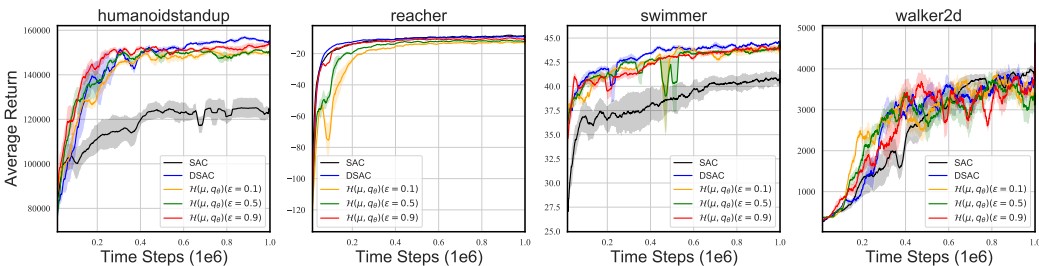

Figure 6: Learning curves of DSAC (C51) with the return distribution decomposition $\mathcal{H}(\mu, q_\theta)$ under different $\varepsilon$.

We study the uncertainty-aware regularization effect from being categorical distributional in the actor-critic framework, where we decompose the C51 critic loss in Distributional SAC (DSAC) according to Eq. 3. We denote the decomposed DSAC (C51) with different $\varepsilon$ as $\mathcal{H}(\mu, q_\theta)(\varepsilon = 0.9/0.5/0.1)$.. As suggested in Figure 6, the performance of $\mathcal{H}(\mu, q_\theta)$ tends to vary from the vanilla DSAC (C51) to SAC with the decreasing of $\varepsilon$ on four MuJoCo environments. In some environments, the difference of $\mathcal{H}(\mu, q_\theta)$ across various $\varepsilon$ may not be pronounced between DSAC (C51) and SAC. We hypothesize that the algorithm performance is not sufficiently sensitive when $\varepsilon$ changes within this restricted range. Although $\varepsilon \in (0, 1)$ is designed to guarantee a valid density decomposition, it does not guarantee that $\epsilon$ in Eq. 3 can flexibly vary from 0 to 1. It is worth noting that our return density decomposition is valid only when $\epsilon \geq 1 - p_E$ as shown in Proposition 1, and therefore $\epsilon$ can not strictly go to 0, where $\mathcal{H}(\mu, q_\theta)$ would degenerate to SAC ideally. Therefore, compared with the ablation study in Figure 3, the trend varying from DSAC to SAC in Figure 6 by decreasing $\varepsilon$ may not be as pronounced as that in value-based RL evaluated on Atari games. One crucial reason behind is that the actor-critic architecture is generally perceived to be more prone to instability compared to value-based learning in RL. As outlined in [15], this instability stems from the policy updates, which likely introduces additional bias or variance from the critic learning process.

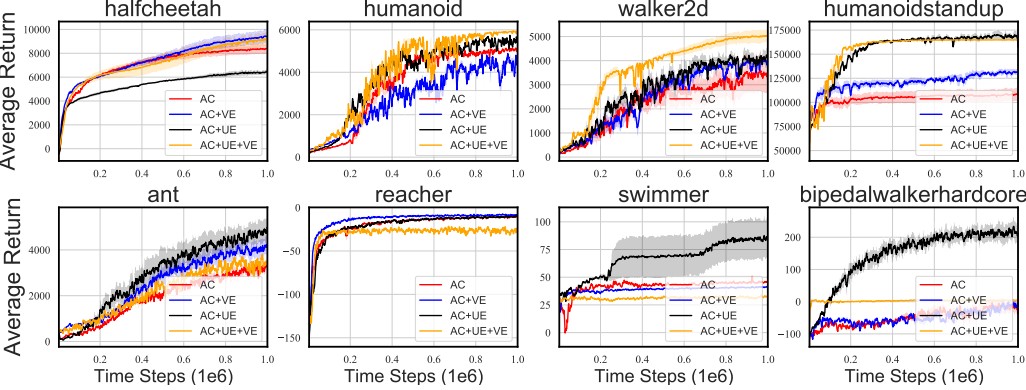

Figure 7: Learning curves of *AC*, *AC+VE* (SAC), **AC+UE** (DAC) and *AC+UE+VE* (DSAC) over five seeds across eight MuJoCo environments where DAC and DSAC are based on IQN. (**First Row**): Mutual improvement. (**Second Row**): Potential interference.

## L.2 Mutual Impacts on DSAC (IQN)

To extend the mutual impact of the two types of regularization to broader distributional RL algorithm, we investigate the learning behavior of distributional RL based on IQN. The conclusion when using IQN is similar to that when using categorical distributional learning in Figure 4. In particular, in the first row of Figure 7, simultaneously employing uncertainty-aware and vanilla entropy regularization renders a mutual improvement. Conversely, the two kinds of regularizations, when adopted together, can also lead to performance degradation, as exhibited in the second row in Figure 7. For instance, on Swimmer and Reacher, *AC+UE+VE* is significantly inferior to **AC+UE** or *AC+VE*. These results about potential interference also serve as the emprical evidence to reveal distinct exploration directions in the policy learning for the two types of regularizations.

## L.3 Ablation Study across Different Bin Sizes (Number of Atoms)

To further demonstrate our regularization effect based on the return density decomposition, we conducted an additional ablation study by varying the number of bins/atoms (equivalent to adjusting the bin sizes) of both C51 and our decompose algorithm $\mathcal{H}(\mu, q_\theta)$. Consistent with the tendency shown in Figure 3 in Section 6.1, Figure 8 also suggests that decreasing $\varepsilon$ implies that $\mathcal{H}(\mu, q_\theta)$ degrades from C51 with the same bin size to DQN. Another interesting observation is that, as shown in Breakout (the first row in Figure 8), increasing the number of atoms (reducing the bin size) restricts the range of $\epsilon$ for a valid return density decomposition in Proposition 1. Consequently, a small number of atoms or a large bin size can allow a broader variation of $\mathcal{H}(\mu, q_\theta)$ from C51 to DQN, facilitating the demonstration of our regularization effect empirically.

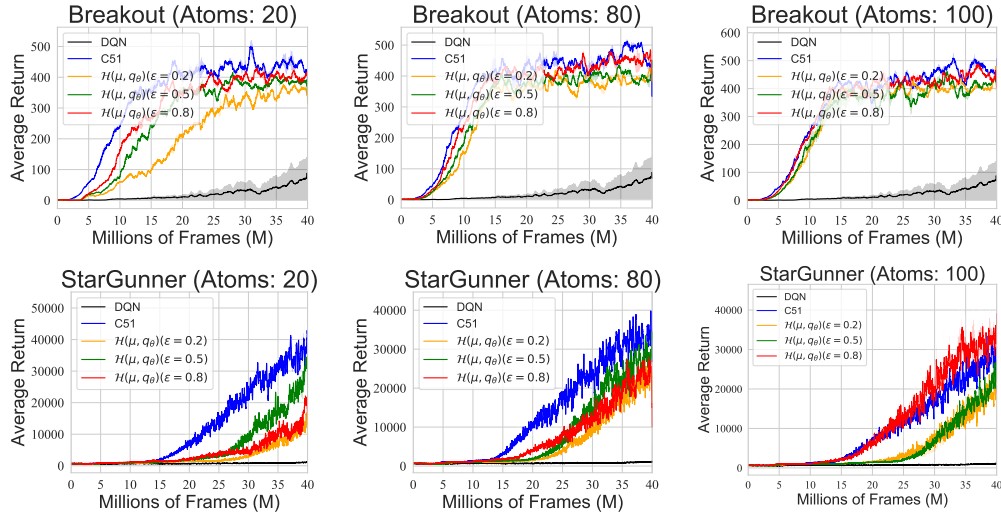

Figure 8: Learning curves of value-based CDRL, i.e., C51 algorithm, and the decomposed algorithm $\mathcal{H}(\mu, q_\theta)$ **across different numbers of atoms (various bin sizes)** on two Atari games. Results are averaged over three seeds, and the shade represents the standard deviation.

# M    Discussion on Decomposing Quantile-based Distributional Loss

In order to extend our analysis and conclusion to broader distributional RL algorithm classes, we need to discuss another commonly-used algorithm based on quantile regression loss [11, 10]. Although it may be possible to discuss both categorical and quantile representation based on the particle representation (Definition 5.13 in [6]), committing either fixed atoms in categorical representation or fixed quantiles in quantile representation can simplify the algorithm analysis. In this section, we discuss how to decompose the quantile-based distributional loss in quantile regression distributional RL.

**Quantile-based Distributional Loss.** In each phase of Neural FZI, we know that the return distribution, parameterized by quantiles, is fixed. This, therefore, leads to a *composite quantile loss* [72], which is initially developed to capture the full conditional distribution of the response variable by predicting or estimating its multiple quantiles:

$$\ell_{\text{quantile}} = \frac{1}{N} \sum_{i=1}^{N} \mathbb{E}_{y \sim P_Y} \left[ \rho_{\tau_i} \left( y - Z_\theta^{\tau_i} \right) \right], \tag{45}$$

where $\rho_{\tau_i}$ is the quantile (pinball) loss defined by $\rho_{\tau_i}(u) = u \left( \tau_i - \mathbb{1}_{\{u<0\}} \right), \forall u \in \mathbb{R}$. We use $P_Y$ to denote the fixed target return distribution. *In quantile-based distributional RL, we can directly sample $y$ from the quantile function $F_Y^{-1}$ of the fixed target return given that both the current and target return distributions are parameterized by the quantiles.* $Z_\theta^{\tau_i}$ represents the estimated $\tau_i$-quantile value of the current return distribution. Alternatively, $\rho_{\tau_i}$ can be the quantile Huber loss [22], a smooth version of vanilla quantile loss at zero, by additionally introducing a hyper-parameter $\kappa$. We thus denote the quantile Huber loss as $\rho_{\tau_i}^\kappa$, which is defined as:

$$\rho_{\tau_i}^\kappa(u) = \left| \tau_i - \mathbb{1}_{\{u<0\}} \right| \frac{\mathcal{L}_\kappa(u)}{\kappa}, \tag{46}$$

where

$$\mathcal{L}_\kappa(u) = \begin{cases} \frac{1}{2} u^2, & \text{if } |u| \leq \kappa \\ \kappa \left( |u| - \frac{1}{2}\kappa \right), & \text{otherwise} \end{cases}. \tag{47}$$

As $\kappa \to 0$, it is easy to show that the quantile Huber loss reverts to the vanilla quantile loss. To simplify the notation, we consider the inner-level loss for a fixed $y$:

$$L_{\text{quantile}} = \frac{1}{N} \sum_{i=1}^{N} \rho_{\tau_i} \left( y - Z_\theta^{\tau_i} \right). \tag{48}$$

**Quantile Representation and Asymptotic Mean-Preserving Property.** The normal representation and categorical representation with the categorical projection $\Pi_\mathcal{C}$ in Eq. 10 could satisfy the mean-preserving property (Section 4.3 in [49]), as seen in (5.18) in Section 5.4 for normal representation [6] and in Lemma 4.8 for categorical representation in [49]. By contrast, quantile distributional dynamic programming is generally not mean-preserving (Lemma 4.8 in [49]), as the quantiles are non-linear functionals of distribution. However, we show that the quantile representation has an asymptotic mean-preserving property as *the mean of quantiles is asymptotically equivalent to the expectation of the considered distribution when the number of quantiles tends to infinity*. Particularly, assume that we have $N$ evenly spaced quantiles $\{ \frac{i}{N+1} \}_{i=1}^{N}$, we approximate the expectation by the mean of all quantiles values defined by

$$\frac{1}{N} \sum_{i=1}^{N} F^{-1}(\frac{i}{N+1}), \tag{49}$$

Consequently, given a random variable $X$ with its quantile function $F^{-1}$, we have the following property of quantile function:

$$\lim_{N \to +\infty} \frac{1}{N} \sum_{i=1}^{N} F^{-1}(\frac{i}{N+1}) = \int_0^1 F^{-1}(\tau) d\tau = \int_{-\infty}^{+\infty} x dF(x) = \mathbb{E}[X], \tag{50}$$

where the first equation results from the relationship between the limit of Riemann Sum and its integral, and the second equation holds by changing the variable $\tau = F(x)$. Note that this asymptotic regime is similar to that in our histogram function analysis for CDRL, where $\Delta \to 0 \iff N \to +\infty$. According to this equivalence regarding the mean quantiles and the expectation of a random variable, we consider the two decomposition ways as follows.

**Decomposition Method 1.** We denote $\bar{Z} = \frac{1}{N} \sum_{i=1}^{N} Z_\theta^{\tau_i}$ as the mean of the quantiles for the *current* return. Consequently, we have a straightforward composition as follows:

$$\rho_{\tau_i}^\kappa \left( y - Z_\theta^{\tau_i} \right) = \rho_{\tau_i}^\kappa \left( (y - \bar{Z}) + (\bar{Z} - Z_\theta^{\tau_i}) \right) = \rho_{\tau_i}^\kappa (y - \bar{Z}) + \delta_\tau, \tag{51}$$

where

$$\delta_\tau = \rho_{\tau_i}^\kappa \left( (y - \bar{Z}) + (\bar{Z} - Z_\theta^{\tau_i}) \right) - \rho_{\tau_i}^\kappa (y - \bar{Z}). \tag{52}$$

Therefore, we have the *decomposed* composite quantile loss as

$$L_{\text{quantile}} = \underbrace{\frac{1}{N} \sum_{i=1}^{N} \rho_{\tau_i}^{\kappa}(y - \bar{Z})}_{\text{Mean-Related Term}} + \underbrace{\frac{1}{N} \sum_{i=1}^{N} \delta_{\tau}}_{\text{Residual Term}}. \tag{53}$$

The first term is a mean-related one, which we will elaborate on later, while the induced $\delta_{\tau}$ in the residual term is aimed at capturing the distribution information beyond only the expectation. Particularly, minimizing $\rho_{\tau_i}^{\kappa}\left((y - \bar{Z}) + (\bar{Z} - Z_{\theta}^{\tau_i})\right)$ in $\delta_{\tau}$ will push the deviations $\bar{Z} - Z_{\theta}^{\tau_i}$ from the current return estimator to capture the deviations from the target return distribution of $y - \bar{Z}$. This regularization term contributes to preserving the richness of the quantile representation for distributional information, especially the dispersion, from the return.

In terms of the mean-related term, let us consider the approximation. As the quantile Huber loss is typically used in quantile-based distributional RL, when $\kappa$ is large, the mean-related term can be simplified as

$$\frac{1}{N} \sum_{i=1}^{N} \rho_{\tau_i}^{\kappa}(y - \bar{Z}) \approx \frac{1}{N} \sum_{i=1}^{N} \left| \tau_i - \mathbb{1}_{\{y - \bar{Z} < 0\}} \right| \frac{1}{2}(y - \bar{Z})^2 \approx \frac{1}{4}(y - \bar{Z})^2, \tag{54}$$

where the first approximation holds because $\mathcal{L}_{\kappa}(u) = \frac{1}{2}u^2$ with high probability. The second approximation holds because $\left| \tau_i - \mathbb{1}_{\{y - \bar{Z} < 0\}} \right| \frac{1}{2}(y - \bar{Z})^2$ is just the quantile value scaled version of least squared loss. Since $\bar{Z}$ is the expectation of all estimated quantiles, it can be approximately symmetric to $\mathbb{E}[Y]$. Suppose $P(y - \bar{Z} < 0) = P(y - \bar{Z} \geq 0) = \frac{1}{2}$, we have

$$\mathbb{E}\left[\left| \tau_i - \mathbb{1}_{\{y - \bar{Z} < 0\}} \right|\right] = \frac{1}{2}\left(|\tau_i - 1| + \tau_i\right) = \frac{1}{2}. \tag{55}$$

Therefore, this approximation in the mean-related term holds, as shown in Eq. 54. This implies that the mean quantile estimator $Z_{\hat{\theta}} = \arg\min_{\bar{Z}} \mathbb{E}_{y \sim P_Y} \left[y - \frac{1}{N}\sum_{i=1}^{N} Z_{\theta}^{\tau_i}\right]^2$ captures the expectation of the target return distribution from $y \sim P_Y$. Recap the asymptotic equivalence between the expected quantiles and the true expectation of a random variable in Eq. 50, the limiting estimator of $Z_{\hat{\theta}}$ by *minimizing the mean-related term in $\ell_{quantile}$* satisfies:

$$\mathbb{E}[Y] = \frac{1}{N} \sum_{i=1}^{N} Z_{\hat{\theta}}^{\tau_i} \to \mathbb{E}\left[Z_{\hat{\theta}}\right] \quad \text{as} \quad N \to +\infty. \tag{56}$$

This implies that the learned expected return $\mathbb{E}\left[Z_{\hat{\theta}}\right]$ is asymptotically mean-preserving when minimizing the mean-related term in the quantile-based distributional loss.

In summary, the first decomposition method decomposes the quantile-base distributional loss into the mean-related and residual terms. After a mild approximation, the mean-related term can be simplified as a least-squared loss equipped with an expected quantiles estimator. Combining the equivalence regarding the limiting behavior of the expected quantiles, *the mean-related term is thus approximately equivalent to the standard least-squared loss used in classical RL, thus asymptotically satisfying the mean-preserving property*. Moreover, the residual term is able to capture the return distribution information beyond its expectation. In the context of uncertain-aware regularized exploration in our paper, the residual term plays a similar role to the cross-entropy-based regularization derived in Proposition 2 of CDRL.

**Decomposition Method 2.** Another decomposition method can directly follow the return density decomposition proposed in Eq. 3, but we apply the decomposition on the quantile function $F^{-1}(\tau)$ for $\tau \in [0, 1]$. We expect that this decomposition also leads to two parts, where the first part can involve the quantile defined on the bin $\Delta_{\bar{\tau}}$ that contains the expected quantiles $\bar{F}^{-1} = \sum_{i=1}^{N} F^{-1}(\tau_i)$, and the second term relates to the distribution part. However, this detailed decomposition is largely beyond the scope of this paper, and it takes more effort to think about it carefully. We leave this decomposition regarding the quantile-based distributional loss as future work.

