# OpenReview forum: "Intrinsic Benefits of Categorical Distributional Loss: Uncertainty-aware Regularized Exploration in Reinforcement Learning"
_NeurIPS.cc/2025/Conference — NeurIPS 2025 poster_

### Official Review · Reviewer_m2h8 · 2025-07-02

**Clarity:** 1
**Significance:** 2
**Originality:** 2
**Rating:** 2
**Confidence:** 3

**Summary:**

The paper aims to explain the benefits of (categorical) distributional RL by relating it to entropy regularization, which encourages exploration. Theoretical results are widely provided, and experiments follow.

**Questions:**

I listed my questions above. My understanding might be inaccurate so please correct me if I was wrong.

**Ethical Concerns:**

["NO or VERY MINOR ethics concerns only"]

**Final Justification:**

i am keeping my score since my concern about the clarity persists after rebuttal.

**Limitations:**

The authors discussed the limitations at the end of the paper. I don't see any potential negative societal impact.

**Paper Formatting Concerns:**

Formatting looks good

**Quality:**

1

**Strengths And Weaknesses:**

This paper presents an interesting perspective on distributional RL by drawing connections to entropy regularization (and thus maxent RL). Both theoretical and experimental components are provided. However, I found the theoretical section somewhat difficult to follow. Some terminology is unclear. For instance, the "Return Density Decomposition" paragraph uses heavy notation and probably need clearer explanations. Terms like $p^\mu_i$, which is the coefficient of the i-th bin, it needs to be strictly defined. The use of the word "induced" in line 161, although it is bold, is kind of clear.

Based on my current understanding, the theoretical contributions may be limited in general. Regarding Proposition 1, I may be misunderstanding it, but it appears to be narrow and not referred to in other theoretical results. Proposition 2 seems fairly standard, as it directly applies the KL divergence to histogram representations. Proposition 3 also appears to be straightforward: when $\Delta$ is approaching 0, the function should reduce to the desired target.

On the empirical side, the paper is verifying that the "uncertainty-aware regularization is crucial to the outperformance of CDRL over classical RL..." (line 335) However I found it difficult to draw this conclusion from the results. According to Figure 3, runs with verifying $\epsilon$ is worse than C51 while being better than DQN. This could suggest that the gains might come from the general benefits of distributional RL rather than the proposed uncertainty-aware regularization itself. Additional analysis or ablation might help clarify the specific contribution of the regularization component.

---

> ### Author Rebuttal · Authors · 2025-07-31
>
> We would like to sincerely express our gratitude to the reviewer for their time and effort in reviewing our paper. We appreciate the reviewer's positive recognition of our work on an interesting perspective on distributional RL, the connection with entropy-regularized RL, and both theoretical and empirical demonstrations. We address each concern you raised and are happy to answer any further questions.
>
> > **Q1:** I found the theoretical section somewhat difficult to follow. Some terminology is unclear. For instance, the "Return Density Decomposition"paragraph uses heavy notation and probably need clearer explanations. Terms like $p_i^\mu$ , which is the coefficient of the i-th bin, it needs to be strictly defined. The use of the word "induced" in line 161, although it is bold, is kind of clear.
>
> **A1:** When it comes to a distribution decomposition, it requires rigorous descriptions and definitions about which random variable of this distribution is, how the support is partitioned, and what the coefficients represent in histogram functions. Although we acknowledge this may increase the cognitive load, we choose this way for the presentation to ensure rigor. For instance, for the induced density function $\hat{\mu}^{s, a}(x)=\sum_{i=1}^N p_i^\mu \mathbb{1}\left(x \in \Delta_i\right) / \Delta$, it rigorously represents a histogram function with each coefficient of this histogram defined on each bin $\Delta_i$. This return distribution decomposition is also illustrated in **Figure 1**, where the coefficient $p_i^\mu$ represents the height on the histogram of the bin $\Delta_i$. **``Induced''** is used intentionally to indicate that given the hyper-parameter $\epsilon$ (under the condition of **Proposition 1**) and a histogram density function $\hat{p}^{s, a}$, $\mu^{s, a}$ results from decomposing the original density function $\hat{p}^{s,a}$, rather than being predefined.
>
> We will revise the text to reduce unnecessary notation and improve explanations of key terms for better clarity and accessibility.
>
>
> > **Q2:** Based on my current understanding, the theoretical contributions may be limited in general. Regarding Proposition 1, I may be misunderstanding it, but it appears to be narrow and not referred to in other theoretical results.
>
>
> **A2:** Thank you for the comment. Firstly, the return distribution decomposition is both novel and non-trivial. It is not generally true but rigorously holds when $\epsilon > 1 -p_E$ as shown in Proposition 1. Proposition 1 serves as the theoretical foundation for the following regularization-based analysis, as the return distribution decomposition will be invalid if the conditions in Proposition 1 are not satisfied. Although Proposition 1 is not directly referenced in subsequent theoretical results, it underpins our analysis with a solid foundation.
>
>
> > **Q3:** Proposition 2 seems fairly standard, as it directly applies the KL divergence to histogram representations.
>
> **A3:** We need to highlight that the tricky part in deriving Proposition 2 is introducing the **return distribution decomposition** to represent the target distribution when we apply the KL divergence, with the detailed proof in **Eq. 28 in Appendix H**. It is also crucial to guarantee a fixed target distribution (ensured by the Neural FZI framework) such that the KL divergence is equivalent to the cross entropy $\mathcal{H}(p, q)$ as shown in **Eq. 27 in Appendix H**. Putting them together, we eventually derive the decomposed Neural FZI objective function in Proposition 2. Importantly, this decomposed Neural FZI objective function is novel, as we first introduce the return distribution decomposition technique that we propose initially in this paper. The derivation avoids complicated assumptions but provides a general and principled formulation when the conditions in Proposition 1 hold.
>
>
> > **Q4:** Proposition 3 also appears to be straightforward: when $\Delta$ is approaching 0, the function should reduce to the desired target.
>
>
> **A4:** Given the novel decomposed objective function of Neural FZI in Proposition 2, it is both novel and non-trivial to analyze the relationship between the minimizer of its first term (i.e., the mean-related component) and the minimizer of the classical RL objective function (i.e., that of Neural FQI). This motivates Proposition 3, whose result was previously unknown before our formulation.
>
> The derivation of Proposition 3 involves a rigorous asymptotic analysis of the Dirac delta function (as a generalized probability density function), an equivalence analysis between distinct minimizers, and properties of the distributional Bellman optimality operator. We emphasize that this derivation is not straightforward, nor can it be directly inferred from existing literature. The conclusion in Proposition 3 is intuitive in hindsight, but this should not be mistaken for simplicity of derivation—it is the result of careful and non-trivial theoretical analysis.
>
>
>
> > **Q5:** On the empirical side, the paper is verifying that the "uncertainty-aware regularization is crucial to the outperformance of CDRL over classical RL..." (line 335) However, I found it difficult to draw this conclusion from the results. According to Figure 3, runs with verifying $\epsilon$ is worse than C51 while being better than DQN. This could suggest that the gains might come from the general benefits of distributional RL rather than the proposed uncertainty-aware regularization itself. Additional analysis or ablation might help clarify the specific contribution of the regularization component.
>
>
> **A5:** We would like to emphasize that the proposed uncertainty-aware regularization is directly derived from the decomposition of the distributional loss function used in CDRL. As shown in **Figure 3**, decreasing $\varepsilon$ from 0.8 to 0.1 reduces the contribution of the distribution component in the baseline algorithm $\mathcal{H}(\mu, q_\theta)$, leading to a learned target $\hat{\mu}^{s,a}$ that captures less high-order moments knowledge of the original return distribution $\hat{p}^{s,a}$ (as defined in Eq. 1). This reduction results in notable performance degradation—from C51 (a distributional RL method) to DQN (a classical RL method)—demonstrating the crucial role of modeling the full return distribution, beyond just its expectation, in enhancing learning performance.
>
> This observed performance gain is rigorously captured by the regularization term in **Eq. 3 of Proposition 2**, which we interpret as uncertainty-aware exploration grounded in the theoretical insights established by our propositions. Accordingly, the ablation study in **Figure 3** precisely illustrates that the decomposed term from the categorical distributional loss plays a pivotal role in explaining the performance gap between classical and distributional RL methods. We welcome any follow-up questions for clarification on this explanation.
>
> Thanks again for the reviewer's consistent dedication to reviewing our work, and we are happy to answer any further questions or concerns. Should this rebuttal address your concerns, we would be grateful for an increased score.

---

> > ### Comment · Area_Chair_Sziz · 2025-08-05
> >
> > Thanks to the authors for their rebuttal! Since this is the only recommendation to reject the papers among the reviewers, a critical evaluation whether the rebuttal addresses the reviewer's criticisms is very important. Could the reviewer please check whether their concerns are alleviated enough to raise their score, or acknowledge even if that is not the case?

---

> > ### Comment · Reviewer_m2h8 · 2025-08-05
> >
> > I appreciate the authors for their detailed response! The explanations are very helpful. However my opinion that the paper is somewhat hard to follow persists. While the authors’ reply changed my understanding of the experiments, which seems to address my concerns about it, I remain quite uncertain due to the lack of understanding of the notations and theory in this paper. After reading authors' response I am still confused about many notations and why they are defined in that way (mostly those surrounding eq 2 and thus those subsequent). I admit that this may be personal, as other reviewers seem not to complaining about the clarity. In summary, I thank the authors again for their response. I will keep my rating and suggest a further polishing (or rewriting notations and some theoretical parts) for the next version.

---

> > > ### Author Response · Authors · 2025-08-06
> > >
> > > We express our gratitude to the reviewer for their continued dedication to reviewing our work. We are happy to hear that our explanations regarding the experiments and the contributions of the propositions were helpful.
> > >
> > > Regarding the reviewer's concern about the clarity and notations, we understand that some parts of our current notation may be dense due to our efforts to maintain formal rigor. We agree that improving clarity in these sections is important, and in the final version, we will revise the notation and clarity and provide more detailed explanations to enhance readability.
> > >
> > > Finally, we would like to express our sincere gratitude for the reviewers’ constructive suggestions and invaluable comments in improving the quality of this work. We extend our deepest respect and appreciation for your thoughtful engagement throughout the review process.

---

### Official Review · Reviewer_WN6r · 2025-07-02

**Clarity:** 4
**Significance:** 4
**Originality:** 4
**Rating:** 5
**Confidence:** 4

**Summary:**

This work revisits the benefits of distributional RL, focusing on categorical distributional RL (CDRL), which uses categorical losses. In Section 4.2, they show that CDRL uses additional information by learning the entire return distribution, compared to standard RL. They also show that the “mean term” in the distributional objective asymptotically behaves like the standard RL objective.

The paper further presents a convergence analysis for distribution-entropy-regularized policy iteration (DERPI), showing that it enjoys similar guarantees as both standard RL and MaxEnt RL. However, they emphasize a key difference: unlike MaxEnt RL, where exploration is explicitly encouraged, DERPI promotes exploration implicitly through the additional reward signal coming from modelling the return distribution, and encouraging the policy to pursue a "target".

The theoretical insights are supported by experiments on both Atari and MuJoCo environments. Also, this work shows that combining both types of regularized exploration (distributional and MatEnt) can sometimes reach better performances.

**Questions:**

1. There are additional parameters involved, e.g. the parameter $\\beta$ in MaxEnt RL. I may have overlooked it, but how is this parameter set in your experiments?

2. Also, when combining the two types of exploration in DSAC, how do you choose the corresponding parameters to balance their contributions? From Figure 4, it seems this combination introduces larger fluctuations in the learning curves. Do you have any insights on whether there are ways to stabilize it?

**Ethical Concerns:**

["NO or VERY MINOR ethics concerns only"]

**Final Justification:**

Most reviewers share similar opinions to the work as I do, and the authors have addressed most of my questions.

**Limitations:**

Yes

**Quality:**

3

**Strengths And Weaknesses:**

## Strengths

This work revisits the benefits of a specific form of distributional RL, i.e., CDRL, and provides a rigorous comparison to standard RL. It clearly shows, by formula, the additional informational used by CDRL. Furthermore, due to the implicit exploration behavior in distributional RL, the paper also theoretical compares to MaxEnt RL. This work makes such comparison clear and also quite elegant.

Overall speaking, this work provides nice insights to the relationships among standard RL, distributional RL, and MaxEnt RL.

Also, the presentation is smooth and well-structured, with the logical flow of ideas.

## Weaknesses

I don’t have much to complain about. However,

1. The analysis is limited to CDRL, rather than general distributional RL frameworks. While this is explicitly acknowledged in the paper, the readers should be careful when intending to generalize the conclusions beyond CDRL.

2. When using UE, the method appears to introduce some instability during training. It could be worth discussing this behavior in more detail.

---

> ### Author Rebuttal · Authors · 2025-07-31
>
> We thank the reviewer for the insightful feedback and positive assessment of our work, especially the clear formula to show the additional information about CDRL and the theoretical comparison with MaxEnt RL. We would like to address the concerns you raised in your review.
>
>
> > **Q1:** The analysis is limited to CDRL, rather than general distributional RL frameworks. While this is explicitly acknowledged in the paper, the readers should be careful when intending to generalize the conclusions beyond CDRL.
>
> **A1:** The main analysis is intended for CDRL, but we hope such a similar conclusion also applies to general distributional RL. An extension to quantile distributional loss is also given in Section 7.
>
> > **Q2:** When using UE, the method appears to introduce some instability during training. It could be worth discussing this behavior in more detail.
>
> **A2:** Thank you for the observation. The instability—such as in the HumanoidStandup task in Figure 4—may result from divergence under certain random seeds. We plan to run additional seeds to better quantify this effect. However, in most environments, introducing UE in algorithms remains stable and does not introduce significant training instability.
>
>
> > **Q3:** There are additional parameters involved, e.g. the parameter $\beta$ in MaxEnt RL. I may have overlooked it, but how is this parameter set in your experiments?
>
> **A3:** Following the original SAC paper, we directly set $\beta=1$ across all experiments.
>
>
> > **Q4:** Also, when combining the two types of exploration in DSAC, how do you choose the corresponding parameters to balance their contributions? From Figure 4, it seems this combination introduces larger fluctuations in the learning curves. Do you have any insights on whether there are ways to stabilize it?
>
> **A4:** We choose the default hyperparameters of distributional RL and MaxEnt RL without tuning them. A key conclusion we make in Section 6.2 is that the combination of the two kinds of entropy can have different mutual impacts, either a mutual improvement (the first row in Figure 4) or potential intervention (the second row in Figure 4). We posit that the potential interference may result from distinct exploration directions in the policy learning for the two types of regularizations, thus potentially leading to a large fluctuation. It is hard to predetermine which exploration is more useful given an environment, and we thus leave it as future work.
>
>
> Thank you again for pointing out these potential areas of improvement. We appreciate your suggestions. Please let us know if you have any further comments or feedback.

---

> > ### Comment · Reviewer_WN6r · 2025-08-06
> >
> > Thank you for your reply. Most of the issues have been resolved. Question 4 remains, but this is fine, as it is perhaps a research question in its own right, and the authors did briefly discuss it in the paper. I have decided to keep my scores.

---

> > > ### Author Response · Authors · 2025-08-06
> > > **Thank you for the positive assessment!**
> > >
> > > We sincerely thank you for your continued positive evaluation and encouragement of our work, which are very helpful. We truly appreciate your warm feedback and extend our deepest respect to you.

---

### Official Review · Reviewer_oCWL · 2025-07-03

**Clarity:** 3
**Significance:** 3
**Originality:** 4
**Rating:** 5
**Confidence:** 4

**Summary:**

The paper attempts to shed light on the improved performance seen in deep RL when using distributional RL, and in particular categorical distributional losses, by decomposing the loss into a mean-related and entropy regularization term. The argument is that this (distribution-matching) entropy regularization term implicitly, but also explicitly in this paper, encourages the policy to visit states/actions where the distribution is poorly estimated.

The primary contributions include this decomposition, theoretical results supporting this decomposition and the correctness/convergence of an explicitly regularized policy evaluation method, empirical evaluation of the explicit form compared with C51 and DQN, and a study in MuJoCo environments on the combination of this form of regularization with the standard form of entropy regularization.

**Questions:**

* On line 239 the text says “we use uppercase notation…” and then proceeds to give an example “(s_t, a_t)” which is lowercase. Uppercase for random variables is ideal, but this juxtaposition was confusing and I thought maybe a typo.
* There are other methods that use categorical losses without attempting to estimate the distribution of returns (at least not fully) such as so called TwoHot encodings and HL-Gauss. Could your approach be adapted to analyze these methods as well? Or in these cases do you think that there would not be a related regularization effect?
* Can you say more about why this form of regularization *should* be expected to be useful, or when it might be counter productive?

**Ethical Concerns:**

["NO or VERY MINOR ethics concerns only"]

**Final Justification:**

Still find this to be strong work worthy of acceptance and author rebuttal did not lower my evaluation. I would be willing to argue in favor of this work.

**Limitations:**

yes

**Quality:**

4

**Strengths And Weaknesses:**

Strengths:

* There is actual novel insight here with rigorous theory and informative experiments to back it up.
* The experimental work asks exactly the right set of questions (for the scope of a conference paper). In Figure 3 we quite clearly see how this explicit form the regularization leads to performance which largely interpolates between C51 and DQN as the strength of regularization is varied. Figure 4 studies a very natural and interesting question of combining with policy entropy regularization, although I feel that this result did not get quite enough discussion for it to fully land.
* The paper is generally well written and the flow of the paper introduces us to this idea very effectively.

Weaknesses:

* The notation is a little rough. It is precise, but there is a fair bit to keep track of and I found it a bit less intuitive than it could be. In particular, I keep wanting to treat mu as the mean-related term and forgetting it is the opposite. Lots of super- and sub-scripts, there’s a lot to keep track of.
* An intuitive explanation of the effects of this form of regularization could be improved. The phrase “lags far behind” seemed like it kept getting used, but I really wanted more investigation and discussion of how to understand the effects here. Overall, I came away from reading this able to parrot that concept but lacking a more complete intuition for what that would really mean in terms of affecting behavior.

---

> ### Author Rebuttal · Authors · 2025-07-31
>
> We thank the reviewer for the insightful feedback and positive assessment of our work, especially on the rigorous theory and experiments to back up our novel insight and effective flow of our paper. We would like to address the concerns you raised in your review.
>
> > **Q1:** The notation is a little rough. It is precise, but there is a fair bit to keep track of and I found it a bit less intuitive than it could be. In particular, I keep wanting to treat mu as the mean-related term and forgetting it is theo pposite. Lots of super- and sub-scripts, there’s a lot to keep track of.
>
> **A1:** Thank you for pointing out this notation issue. We will simplify notations and improve clarity to enhance the readability in the final version of the paper.
>
> > **Q2:** An intuitive explanation of the effects of this form of regularization could be improved. The phrase “lags far behind” seemed like it kept getting used, but I really wanted more investigation and discussion of how to understand the effects here. Overall, I came away from reading this able to parrot that concept but lacking a more complete intuition for what that would really mean in terms of affecting behavior
>
> **A2:** When revisiting the objective function in **Eq. 8 of Lemma 1**, optimizing the policy implies an increase of the regularization term $f(\mathcal{H} ( \mu^{\mathbf{s}\_t, \mathbf{a}\_t},  q_{\theta}^{\mathbf{s}\_t, \mathbf{a}\_t} ) )$. Given a fixed $q_\theta^{s,a}$, the increasing of $f(\mathcal{H} ( \mu^{\mathbf{s}\_t, \mathbf{a}\_t},  q_{\theta}^{\mathbf{s}\_t, \mathbf{a}\_t} ))$ indicates that the future trajectory with the states and actions is expected to have a large distribution discrepancy. Given the current state, the policy is optimized to explore states and actions whose current return distribution $q_\theta^{s, a}$ is dramatically different from the estimated target return distribution $\mu^{s, a}$. Intuitively, the exploration is promoted to diminish the agent's current understanding of the environmental uncertainty (through the return distribution) and the (estimated) real environmental uncertainty. We will add a more detailed explanation in the revised version.
>
>
>
> > **Q3:** On line 239 the text says “we use uppercase notation…” and then proceeds to give an example “(s_t, a_t)” which is lowercase. Uppercase for random variables is ideal, but this juxtaposition was confusing and I thought maybe a typo.
>
> **A3:** Thank you for catching this typo. The notation should use boldface to represent random variables, not uppercase letters. This follows the convention used in much of the RL literature, such as in Soft Actor-Critic. We will correct the text for clarity in the final version.
>
>
> > **Q4:** There are other methods that use categorical losses without attempting to estimate the distribution of returns (at least not fully) such as so called TwoHot encodings and HL-Gauss. Could your approach be adapted to analyze these methods as well? Or in these cases do you think that there would not be a related regularization effect?
>
> **A4:** This is a great question. It is indeed possible to extend our analysis to methods like HL-Gauss [1, 2], which also utilize histogram-like representations, albeit under a Gaussian assumption. Our histogram-based return distribution decomposition can be potentially applied to HL-Gauss and we expect that a similar regularization effect may emerge from their simplified formulations. We consider this an interesting direction for future work.
>
>
> > **Q5:** Can you say more about why this form of regularization should be expected to be useful, or when it might be counter productive?
>
> **A5:** This regularization promotes exploration by encouraging the agent to reduce the environmental uncertainty between the estimated and target return distributions by assuming that **the target distribution is well-estimated**. However, if the TD target is poorly estimated (e.g., due to limited distributional representation or the limitations of one-step TD as opposed to multi-step TD), the policy may be misled toward uninformative regions of the state space. In such cases, the distributional loss can be counterproductive, and classical RL may perform better.
>
>
> Thank you again for pointing out these potential areas of improvement. We appreciate your suggestions. Please let us know if you have any further comments or feedback.
>
> ### Reference
>
> [1] Stop Regressing: Training Value Functions via Classification for Scalable Deep RL
> [2] Improving regression performance with distributional losses. (ICML 2018)

---

### Official Review · Reviewer_5szr · 2025-07-03

**Clarity:** 2
**Significance:** 3
**Originality:** 4
**Rating:** 5
**Confidence:** 4

**Summary:**

This paper seeks to deepen the understanding of the benefits of the Categorical Distributional Loss. It does so by decomposing the return distribution estimated in distributional RL into the Dirac delta distribution of the mean and the remaining distribution capturing higher-moments information, and then applying the distributional analogue of the neural fitted Q iteration to analyze the objective that categorical distributional RL is minimizing.

This decomposition and analysis uncovers a connection between distributional RL, whose objective seeks to reach states where the cross-entropy between the estimation of the distribution at the current state and the distribution at the target state is more, and maximum entropy (MaxEnt) RL, whose objective aims to reach states where the entropy of the policy is maximized.

Experiments in Atari and MuJoCo seek to validate this analysis. First, by showing that the performance of Categorical DQN can be smoothly reduced to DQN by varying how much the algorithm focuses on minimizing the cross-entropy term. Second, by comparing exploration caused by the cross-entropy minimization versus the action entropy maximization.

**Questions:**

The last two subsections of Section 5.1 indicate that the policy seeks states that have a higher discrepancy between the distribution estimate at that state and the return distribution from the next states. However it is not made clear how the policy is affected over predecessor states where this discrepancy might not have propagated yet, and also how that discrepancy might propagate back. It is not simply a question of comparing to MaxEnt RL since there the entropy regularization is not an error term.

**Ethical Concerns:**

["NO or VERY MINOR ethics concerns only"]

**Final Justification:**

Authors addressed my minor concerns through rebuttal. I do not feel a need to update my rating

**Limitations:**

Yes

**Quality:**

4

**Strengths And Weaknesses:**

## Strengths
* The analysis of the paper seems solid
* The appendix includes a lot of details and is well constructed.
* The experiments are chosen well to validate the hypothesis

## Weaknesses
* The way the idea is communicated can be improved further. The core result of the paper is hard to parse unless the reader is familiar with a lot of the related literature. But given the deep analysis of this paper, perhaps that is the audience they are tailoring it for.
* The experiments on the Atari domain should be replicated in MuJoCo if possible to showcase that the idea holds in discrete and continuous action spaces as well as with value based and actor critic based algorithms.
* The results in the bottom row of Figure 4 are mixed, but the paper merely presents a hypothesis for the mixed results instead of validating the hypothesis.
* Minor point: Evaluations with 3 seeds in Atari and 5 seeds in MuJoCo might not be statistically sound, but consistency across the different tasks in these two domains points to some validity.
* Minor point, another paper that has used a mixture of distributional RL and MaxEnt RL to great success is [1]. Perhaps that paper validates or disproves some of the hypotheses in this one.

### References
[1] Outracing champion Gran Turismo drivers with deep reinforcement learning., Wurman et al., Nature, 2022

---

> ### Author Rebuttal · Authors · 2025-07-31
>
> We thank the reviewer for the insightful feedback and positive assessment of our work, especially on solid analysis, detailed appendix, and well chosen experiments. We would like to address the concerns you raised in your review.
>
>
> > **Q1:[Weakness 1]** The way the idea is communicated can be improved further. The core result of the paper is hard to parse unless the reader is familiar with a lot of the related literature. But given the deep analysis of this paper, perhaps that is the audience they are tailoring it for.
>
> **A1:** Thank you for the helpful feedback. In future revisions, we will improve the clarity and presentation of the core ideas to make them more accessible to a broader audience while still retaining the depth of our analysis.
>
> >  **Q2:[Weakness 2]** The experiments on the Atari domain should be replicated in MuJoCo if possible to showcase that the idea holds in discrete and continuous action spaces as well as with value based and actor critic based algorithms.
>
> **A2:** We have conducted additional experiments in MuJoCo environments, and similar results can be found in **Appendix M.1**.  This empirical evidence consistently demonstrates the key effect of the decomposed regularization term from the categorical distribution loss on the superiority of and distributional RL over classical RL.
>
>
> >  **Q3:[Weakness 3]** The results in the bottom row of Figure 4 are mixed, but the paper merely presents a hypothesis for the mixed results instead of validating the hypothesis.
>
> **A3:** Yes. Results in **the first row of Figure 4** show that simultaneously employing uncertainty-aware and vanilla entropy regularization renders a mutual improvement. By contrast, the two kinds of regularization can also lead to potential interference in policy optimization through the results in **the second row of Figure 4**, revealing an interesting phenomenon to be explored in the future.
>
>
>
> >  **Q4:[Weakness 4]** Minor point: Evaluations with 3 seeds in Atari and 5 seeds in MuJoCo might not be statistically sound, but consistency across the different tasks in these two domains points to some validity.
>
> **A4:** Thank you for this suggestion, and we will run more seeds in the final version of the paper.
>
>
> >  **Q5:[Weakness 5]** Minor point, another paper that has used a mixture of distributional RL and MaxEnt RL to great success is [1]. Perhaps that paper validates or disproves some of the hypotheses in this one.
>
> **A5:** Thanks for sharing this reference paper, which indeed validates that combining distribution learning and MaxEnt RL can lead to mutual improvement in some scenarios.
>
>
> > **Q6:[Questions]** The last two subsections of Section 5.1 indicate that the policy seeks states that have a higher discrepancy between the distribution estimate at that state and the return distribution from the next states. However, it is not made clear how the policy is affected over predecessor states where this discrepancy might not have propagated yet, and also how that discrepancy might propagate back. It is not simply a question of comparing to MaxEnt RL since there the entropy regularization is not an error term.
>
> **A6:** Thank you for the insightful question. It is correct that the policy is encouraged to explore states with high discrepancy between the current estimated return distribution and the target return distribution. Since the policy operates in an MDP, this exploratory behavior is primarily driven by the current state rather than directly by predecessor states.
>
> However, we agree that the influence of discrepancies in predecessor states can indirectly propagate backward through the Bellman updates during training. If the discrepancy in the predecessor states may not propagate yet, the policy may not be immediately simulated to visit those earlier states, which could limit the full exploitation of distributional uncertainty in long trajectories. This subtle effect on the trajectory-based exploration is beyond the scope of this paper, but it is interesting to have a deeper theoretical and empirical investigation. We will consider this in future work.
>
> Thank you again for pointing out these potential areas of improvement. We appreciate your suggestions. Please let us know if you have any further comments or feedback.

---

> > ### Comment · Reviewer_5szr · 2025-08-05
> > **Response to Author Rebuttal**
> >
> > Thank you for the comments on the review
> >
> > > A2: We have conducted additional experiments in MuJoCo environments, and similar results can be found in Appendix M.1. This empirical evidence consistently demonstrates the key effect of the decomposed regularization term from the categorical distribution loss on the superiority of and distributional RL over classical RL.
> >
> > Ah I missed those results at the end of the appendix. These alleviate one of my main concerns
> >
> > As for the responses to the other points raised, I acknowledge the authors' rebuttal that most of these would be useful to address in future work.

---

> > > ### Author Response · Authors · 2025-08-06
> > >
> > > We are glad that our responses are helpful to alleviate the reviewer's main concerns. We're also grateful that the reviewer recognizes the potential value of addressing the remaining points in future work. We will make sure to clarify these aspects more explicitly in the camera-ready version to avoid any ambiguity. Thank you so much!

---

### Note · Authors · 2025-08-13

Dear all reviewers and ACs,


To make full use of the final remark, we again sincerely thank all reviewers for their time, detailed feedback, and thoughtful engagement with our paper. We appreciate that multiple reviewers recognized our solid analysis, a novel decomposition of the distributional loss, new theoretical insights into uncertainty-aware regularization, detailed descriptions with an effective paper flow, and well-chosen experiments across both discrete and continuous action spaces.

We also appreciate Reviewer m2h8's feedback on clarity. We understand that some of our current notations may have been dense due to our efforts to maintain formal rigor. We will incorporate these suggestions into the final version with improved readability and additional discussion.

We hope the revisions and our responses address the main concerns and that the paper can contribute meaningfully to the community — especially by clarifying how distributional perspectives can benefit RL and sequential decision-making research more broadly.

Sincerely,
Authors

---

### Decision · Program_Chairs · 2025-09-17

**Decision:**

Accept (poster)

**Comment:**

This paper is divisive among reviewers. Everyone appreciates the interesting insightful theoretical perspective on an old empirical observation that mystifies researchers for almost a decade now. However, the paper has two critical flaws: 1. the analysis is complex, hard to understand without extensive prior knowledge and uses overly complex notation; and 2. the empirical results show the effect in some experiments, but remain ambiguities in others.

A clear majority of reviewers, who state they are also more confident in their assessment, accepts the paper is for a more niece audience that is fluent enough in literature and willing to disentangle the complex notation. Only m2h8 maintains that the paper is too hard to read and the interpretation of the results questionable. This could be easily dismissed (m2h8 does so as well) as an individual opinion, but I need to admit that I somewhat agree. Complexity is relative, but the interpretation of the results is not as clean cut as the authors make it seem. For example, Figure 8 in the appendix ablates the influence of the proposed regularizer for Mujoco environments. However, here the results do not "interpolate" as they do for Atari environments in Figure 3. These kinds of ambiguities are common in deep RL research, but they do cast doubt on some of the empirical conclusions.

I nonetheless recommend to accept this paper. The theory, while hard to digest, sheds light on an old unresolved question, and the results provide enough evidence to make the authors' interpretation reasonable, if not unquestionable.